# Antibodies to repeat-containing antigens in *Plasmodium falciparum* are exposure-dependent and short-lived in children in natural malaria infections

Madhura Raghavan[1], Katrina L Kalantar[2], Elias Duarte[3], Noam Teyssier[1], Saki Takahashi[1], Andrew F Kung[1], Jayant V Rajan[1], John Rek[4], Kevin KA Tetteh[5], Chris Drakeley[5], Isaac Ssewanyana[4,5], Isabel Rodriguez-Barraquer[1,6], Bryan Greenhouse[1,6]*, Joseph L DeRisi[1,6]*

[1]University of California, San Francisco, San Francisco, United States; [2]Chan Zuckerberg Initiative, Redwood City, United States; [3]University of California, Berkeley, Berkeley, United States; [4]Infectious Diseases Research Collaboration, Kampala, Uganda; [5]London School of Hygiene and Tropical Medicine, London, United Kingdom; [6]Chan Zuckerberg Biohub, San Francisco, United States

*For correspondence:
bryan.greenhouse@ucsf.edu
(BG);
joe@derisilab.ucsf.edu (JLDeR)

**Abstract** Protection against *Plasmodium falciparum*, which is primarily antibody-mediated, requires recurrent exposure to develop. The study of both naturally acquired limited immunity and vaccine induced protection against malaria remains critical for ongoing eradication efforts. Towards this goal, we deployed a customized *P. falciparum* PhIP-seq T7 phage display library containing 238,068 tiled 62-amino acid peptides, covering all known coding regions, including antigenic variants, to systematically profile antibody targets in 198 Ugandan children and adults from high and moderate transmission settings. Repeat elements – short amino acid sequences repeated within a protein – were significantly enriched in antibody targets. While breadth of responses to repeat-containing peptides was twofold higher in children living in the high versus moderate exposure setting, no such differences were observed for peptides without repeats, suggesting that antibody responses to repeat-containing regions may be more exposure dependent and/or less durable in children than responses to regions without repeats. Additionally, short motifs associated with seroreactivity were extensively shared among hundreds of antigens, potentially representing cross-reactive epitopes. PfEMP1 shared motifs with the greatest number of other antigens, partly driven by the diversity of PfEMP1 sequences. These data suggest that the large number of repeat elements and potential cross-reactive epitopes found within antigenic regions of *P. falciparum* could contribute to the inefficient nature of malaria immunity.

## Editor's evaluation

Malaria immunity is complex, and this new platform, namely the phage display of *Plasmodium falciparum* proteome-wide peptides for profiling of antibody targets, provides a valuable addition to the toolkit for understanding humoral responses. The study, conducted using plasma from Ugandan children and adults, represents an important aspect of naturally acquired antibodies with sera-reactive responses to the intra-and inter-protein repeat regions. The revised version solidly supports the claims of the authors; it contains a reanalysis of cohort comparisons accounting for infection status, updated analyses of cross-reactive epitopes to account for within-individual effects, and it emphasizes the limitations in the conclusions.

## Introduction

Malaria, a disease caused by the single-celled eukaryotic parasite *Plasmodium*, caused an estimated 241 million cases and 627,000 deaths in 2020, mostly by the species *Plasmodium falciparum (P. falciparum;* WHO Report 2021). Various strategies have been adopted for elimination of malaria, focusing on vector control, chemoprevention and vaccines. In 2021, the World Health Organization (WHO) made its first recommendation for widespread use of a malaria vaccine, RTS,S. While this is an encouraging step, there is nevertheless need for improvement as the efficacy of RTS,S is only 30–40% and protection wanes in a few months despite a four-dose regimen (*Clinical Trials Partnership, 2015*; *Olotu et al., 2016*). To design more effective vaccines, a deeper understanding of the nature of acquired immunity to malaria is critical.

Natural protection against malaria, particularly protection from uncomplicated malaria and ability to control parasitemia, requires multiple exposures and wanes upon cessation of exposure (*Doolan et al., 2009*). This naturally acquired immunity develops gradually with age and increasing cumulative exposure to *P. falciparum* in endemic settings, where adults may obtain substantial protection from disease, and children under 5 face the highest risk of death (*Doolan et al., 2009*; WHO Report 2020). While a comprehensive understanding of the factors influencing the slow development of immunity is still lacking, certain properties of parasite antigens have been proposed to contribute (*Portugal et al., 2013*). These include, amongst other properties, antigenic variation, antigens containing repeat elements and cross-reactive epitopes (*Anders, 1986*; *Reeder and Brown, 1996*; *Schofield, 1991*). While antigenic variation has been extensively studied in malaria, a systematic investigation of repeat-containing antigens and cross-reactive epitopes has been lacking.

Repeat elements are those where identical or similar motifs are repeated in tandem or with spaces within a protein. Repeat elements are widely prevalent in the proteome of *P. falciparum* and have been described to be highly immunogenic in a few antigens (*Davies et al., 2017*), such as the short, linear 'NANP' repeats from circumsporozoite protein (CSP) present in the RTS,S and R21 vaccines (*Cockburn and Seder, 2018*). Due to increased valency, epitopes in repeat elements can behave differently in comparison to the presentation of the same epitope as a single copy and have the potential to alter the nature of the resulting response. For instance, increased valency may lead to increased plasmablast formation by increasing the strength of the antigen-B-cell receptor (BCR) interaction, potentially altering the T-dependent response and inducing a T-independent response (*Feldmann and Easten, 1971*; *Kato et al., 2020*; *O'Connor et al., 2006*; *Ochiai et al., 2013*; *Paus et al., 2006*; *Schofield, 1991*; *Schwickert et al., 2011*). Although a few repeat antigens in *P. falciparum* have been well characterized, there has not been a comprehensive investigation of repeat elements with respect to their seroreactivity and their associations with humoral development.

The presence of biochemically similar epitopes can lead to cross-reactivity with antibodies and BCR. While non-identical repeat elements may represent such potential cross-reactive epitopes within a protein, similar epitopes may also be present across different proteins. How the quality of humoral response may be impacted by the presence of cross-reactive epitopes remains largely unexplored, although a study with viral variant antigens points to a frustrated affinity maturation process due to conflicting selection forces from variant epitopes (*Wang et al., 2015*). A handful of cross-reactive epitopes have been reported in *P. falciparum* (*Wåhlin et al., 1992*) and have been proposed to negatively impact the affinity maturation process, although direct evidence is lacking (*Anders, 1986*). To obtain a deeper understanding of how cross-reactive epitopes influence B cell immunity to malaria, a comprehensive atlas of cross-reactive epitopes across the *P. falciparum* proteome is first needed.

A systematic proteome-wide investigation of the humoral response to *P. falciparum* would provide important insights to our understanding of malaria immunity, including features such as repeat elements and cross-reactive epitopes. Specific technical challenges have impeded progress in this area. Although high-throughput approaches like protein arrays and alpha screens have reached a high coverage of the *P. falciparum* proteome (*Camponovo et al., 2020*; *Morita et al., 2017*), they do not allow for high-resolution, characterization of regions within antigenic proteins. In contrast, peptide arrays offer high-resolution antigenic profiling but are inherently limited to the numbers of targets that can be produced and printed on an array, usually in the order of tens of proteins (*Hou et al., 2020*; *Jaenisch et al., 2019*).

Here, we customized a programmable phage display system (PhIP-seq; *Larman et al., 2011*), previously used for antigenic profiling in many diseases, including autoimmune disorders and viral

**Table 1.** Characteristics of the Ugandan cohorts.

| Region | Age group (yrs) | No. of people | Proportion positive for infection at the time of sample collection | Time since last infection (days) - *median (IQR)* | Incidence of symptomatic malaria per year - *median (IQR)* | Household annual EIR* (infective bites / person) - *median (IQR)* |
|---|---|---|---|---|---|---|
| | 2–3 | 10 | 0.5 | 18.5 (0,85) | 5.8 (2.9,7.7) | 56 (33,148) |
| | 4–6 | 30 | 0.66 | 0 (0,45) | 3.6 (2.6,4.8) | 59 (38,84) |
| | 7–11 | 30 | 0.63 | 0 (0,45) | 2.3 (2,4.3) | 46 (30,110) |
| Tororo | >18 | 30 | 0.7 | 0 (0,45) | 1.2 (0.9,1.6) | 49 (35,94) |
| | 2–3 | 10 | 0.1 | 155 (61,190) | 1.7 (0.9,2) | 4.3 (4, 14) |
| | 4–6 | 30 | 0.2 | 114 (43,289) | 1.5 (0.7, 2.3) | 7.3 (4.5, 15) |
| | 7–11 | 30 | 0.13 | 121 (41,263) | 1.5 (0.6, 2) | 5.2 (4, 7) |
| Kanungu | >18 | 30 | 0.2 | 109 (38, 223) | 1.1 (0.8, 1.3) | 6.8 (4.8, 15.4) |

*EIR – Entomological Inoculation Rate.

infections (*Mandel-Brehm et al., 2019*; *Rajan et al., 2021*; *Vazquez et al., 2020*; *Zamecnik et al., 2020*), for interrogation of the humoral response to *P. falciparum* infection. We designed a custom library ('Falciparome') that features over 238,000 individual 62 amino acid peptides encoded in T7 Phage, tiled every 25 amino acids across all annotated *P. falciparum* open reading frames from 3D7/ IT genomes with additional variant antigenic sequences. Importantly, PhIP-seq leverages advances in next-generation sequencing to effectively convert serological assays into digital sequence counts. Furthermore, programmable phage display allows iterative enrichment, driving a high signal to background ratio with high specificity and sensitivity (*O'Donovan et al., 2020*).

We performed PhIP-seq with the Falciparome phage library to characterize the targets of the naturally acquired antibody response to *P. falciparum* in high-resolution, leveraging well-defined cohorts composed of 198 Ugandan children and adults from two different transmission settings and compared these to a large set of US anonymous blood donors. The resulting high-resolution atlas of seroreactive peptides suggest that antibody responses to repeat-containing regions are more exposure-dependent and/or less durable in children, compared to antibody responses to regions without repeats. Further, an extensive presence of potential cross-reactive motifs was identified among antigenic peptides from many proteins highly targeted by the immune system. These results have important implications for understanding the nature of humoral response in malaria and the future vaccine designs.

## Results

PhIP-seq was performed on plasma samples selected from two Ugandan cohorts with household level data on entomologic exposure as well as detailed individual characteristics (*Table 1*). For this study, we selected a single sample from each of 200 age-stratified individuals (children aged 2–11 years and adults) from two different sites in Uganda: Tororo, a region which had very high malaria transmission at the time of sampling (annual entomological inoculation rate [EIR] - 49 infective bites per person) and Kanungu, a region of moderately high transmission (EIR - 5 infective bites per person; *Kamya et al., 2015*). While the majority of individuals from Tororo were positive for infection at the time of sampling, those from Kanungu were sampled at a median of 100 days after their previous infection. We have previously shown that children acquire clinical immunity to malaria more rapidly in Tororo than Kanungu, consistent with higher rates of exposure (*Rodriguez-Barraquer et al., 2018*), and that adults at both sites have substantial immunity (*Rek et al., 2016*).

### Falciparome library design

We constructed a T7 phage display library programmed to display the entire proteome of *P. falciparum* in 62-amino acid peptides with 25-amino acid step size, resulting in 37-amino acid tiling, referred to as the Falciparome. (Methods, *Figure 1*). The complete design files are available at Dryad doi:10.7272/

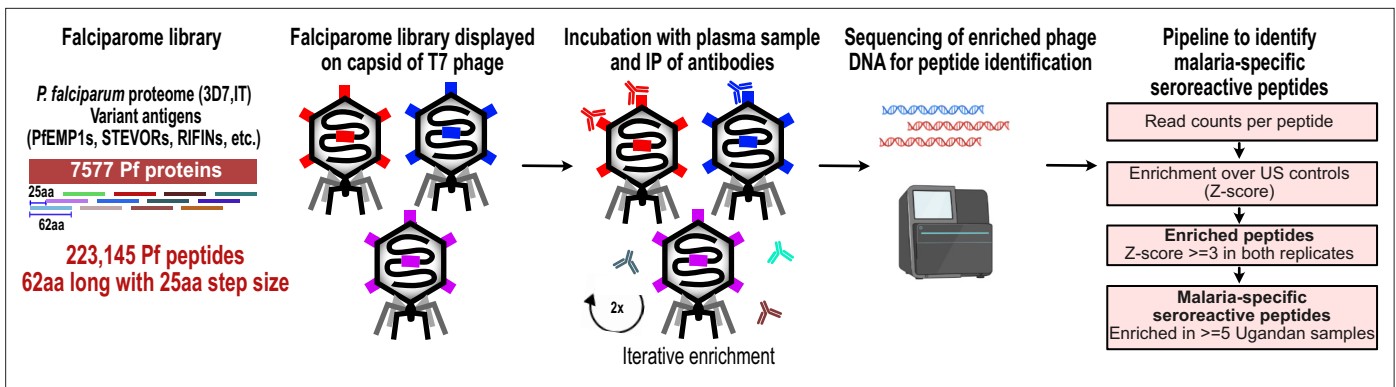

**Figure 1.** PhIP-seq overview and analysis pipeline. Falciparome phage library displays the proteome of *Plasmodium falciparum* in 62-aa peptides with 25-aa step size on T7 phage and also includes variant sequences of many antigens, including major Variant Surface Antigens (VSA). PhIP-seq was performed with incubation of Falciparome library with human plasma, followed by IP of antibodies in the sample and enrichment of antibody binding phage. Two rounds of enrichment were performed and enriched phage were sequenced to obtain the identity of the encoded peptides. A filtering pipeline was then used to identify seroreactive peptides specific to the malaria cohort.

The online version of this article includes the following figure supplement(s) for figure 1:

**Figure supplement 1.** Pipeline for library construction.

Q69S1P9G and protocol at 10.17504/protocols.io.j8nlkkrr5l5r/v1. Overall, the library includes 238,068 peptides from 8980 protein sequences, including all known protein sequences from 2 reference strains (3D7 and IT) and extensive diversity of variant sequences from key antigens including PfEMP1s, RIFINs and STEVORs (*Table 2*, *Figure 1—figure supplement 1*, Methods). Of these, 223,145 peptides from 7577 proteins are from *P. falciparum*. Greater than 99.5% of the programmed peptides were represented in the final packaged phage library with relatively uniform distribution of abundance, with 90% of the peptides within a 16-fold difference (*Figure 2—figure supplement 1*).

## PhIP-seq using the Falciparome library robustly identifies peptides that differentiate individuals in malaria endemic areas from US controls

PhIP-seq was performed with Falciparome on less than 1 μl plasma from the 200 Ugandan cohort samples, and 86 samples from New York Blood Center (US controls) were run for non-specific background correction, assuming most were unlikely to have been exposed. Two rounds of enrichment were performed. The scalability of the assay allowed for high-throughput processing of all 286 samples in replicates. High correlation observed between technical replicates (Pearson r: median (IQR)=0.96 (0.92–0.98)) indicated that the technique was highly reproducible (*Figure 2—figure supplement 2*). Prior to any filtering, a clean separation of Ugandan and US controls was observed (*Figure 2—figure supplement 2*). Furthermore, expected target peptides were enriched in a sample-specific manner - PhIP-seq with a polyclonal control antibody α-GFAP highly enriched for GFAP peptides and seroreactivity against a common virus, Epstein-Barr virus (EBV), was higher across all human samples than in the control α-GFAP experiment (*Figure 2—figure supplement 3*). Two Ugandan samples were dropped due to low quality data, resulting in 198 Ugandan samples for further analysis.

A stringent analysis pipeline was implemented to identify malaria-specific enriched peptides while minimizing the potential for false positives. An increase in base read counts (enrichment) compared to US controls (Z-score ≥ 3 in both replicates in a given sample) was implemented, plus a requirement that the enrichment be present in at least five Ugandan samples (Materials and methods, *Figure 1*, *Figure 2—figure supplement 4*) to identify malaria-specific enriched peptides ('seroreactive peptides'). Using this conservative approach, a total of 9927 peptides were identified as seroreactive across all samples, representing the identified targets of antibodies in this study (*Supplementary file 1*).

**Table 2.** Composition of Falciparome phage library.

| | Input sequences before collapsing on similarity | Identity threshold for collapsing by CD-HIT | # Final collapsed Protein sequences |
|---|---|---|---|
| *P. falciparum* reference proteome | 3D7, IT (10,771 total) | 99% | 6372 |
| *P. falciparum* variant sequences | • PfEMP1 (431 from 3D7, IT, IGH, RAJ116, PFCLIN, IT4, DD2 genomes)<br>• RIFIN (all 3D7+IT)<br>• STEVOR (all 3D7+IT)<br>• SURFIN (all 3D7+IT + 15)<br>• AMA1 (2)<br>• CSP (6)<br>• MSPDBL1 (6)<br>• MSPDBL2 (5)<br>• PfMC2TM (all 3D7+IT) | | |
| *Other variants* | *P. reichnowi PfEMP1 (PFREICH)*<br>*Anopheles* - CE5 (5), SG6 (5) | 100%<br>(90% for CSP) | 1205 |
| *Anopheles* salivary proteins | 53 proteins from 19 Anopheles species as described in Figure 1 of *Arcà et al., 2017* | 98% | 708 |
| Vaccine/Viral/Toxin sequences | • Tetanus<br>• Diphtheria<br>• Pertussis<br>• EBV<br>• Measles<br>• Mumps<br>• Rubella<br>• Polio<br>• RotoAB | 98%<br>(90% for RotoAB) | 684 |
| Laboratory positive controls | • GFAP<br>• GFP<br>• Gephryn<br>• MYC, NR1<br>• Tubulin (alpha/beta) | 98% | 11 |
| TOTAL PROTEINS | | | 8,980 |
| TOTAL PEPTIDES | | | 238,068 |

## Overview of the malaria-specific seroreactive peptides identified with PhIP-seq

The 9927 seroreactive peptides identified by the pipeline were derived from 1648 parasite proteins ('seroreactive proteins') and antigenic variants, representing approximately 30% of the 5400 member proteome of *P. falciparum*, many of which showed broad seroreactivity across pediatric and adult Ugandan samples, whereas these same peptides showed no or rare seroreactivity in US controls (*Figure 2a*). The number of peptides enriched ('breadth') in children from moderate transmission settings was less than half of that in children in high transmission settings or adults in either setting (*Figure 2b*), an observation that we examined in greater detail below.

The 1648 seroreactive proteins identified here have reported expression across the lifecycle stages occurring in the human – sporozoite, asexual, and sexual blood stages (*Figure 2c*). Although liver stage proteomic *P. falciparum* datasets were not available for comparison, several known liver stage antigens in the dataset (LSA1, LSA3, etc.) were detected. Notably, very few of the proteins were exclusively detected in the mosquito oocyst stage. Among the 40 seroreactive proteins with the highest seropositivity (percent of people enriching for at least one seroreactive peptide in that protein), protective antibodies have been reported for 20 of them (*Supplementary file 2*). Moreover, as expected, and consistent with previous studies, the top seroreactive proteins (*Supplementary file 3*) were enriched for those at the host-parasite interface (GO analysis – *Figure 2—source data 1*).

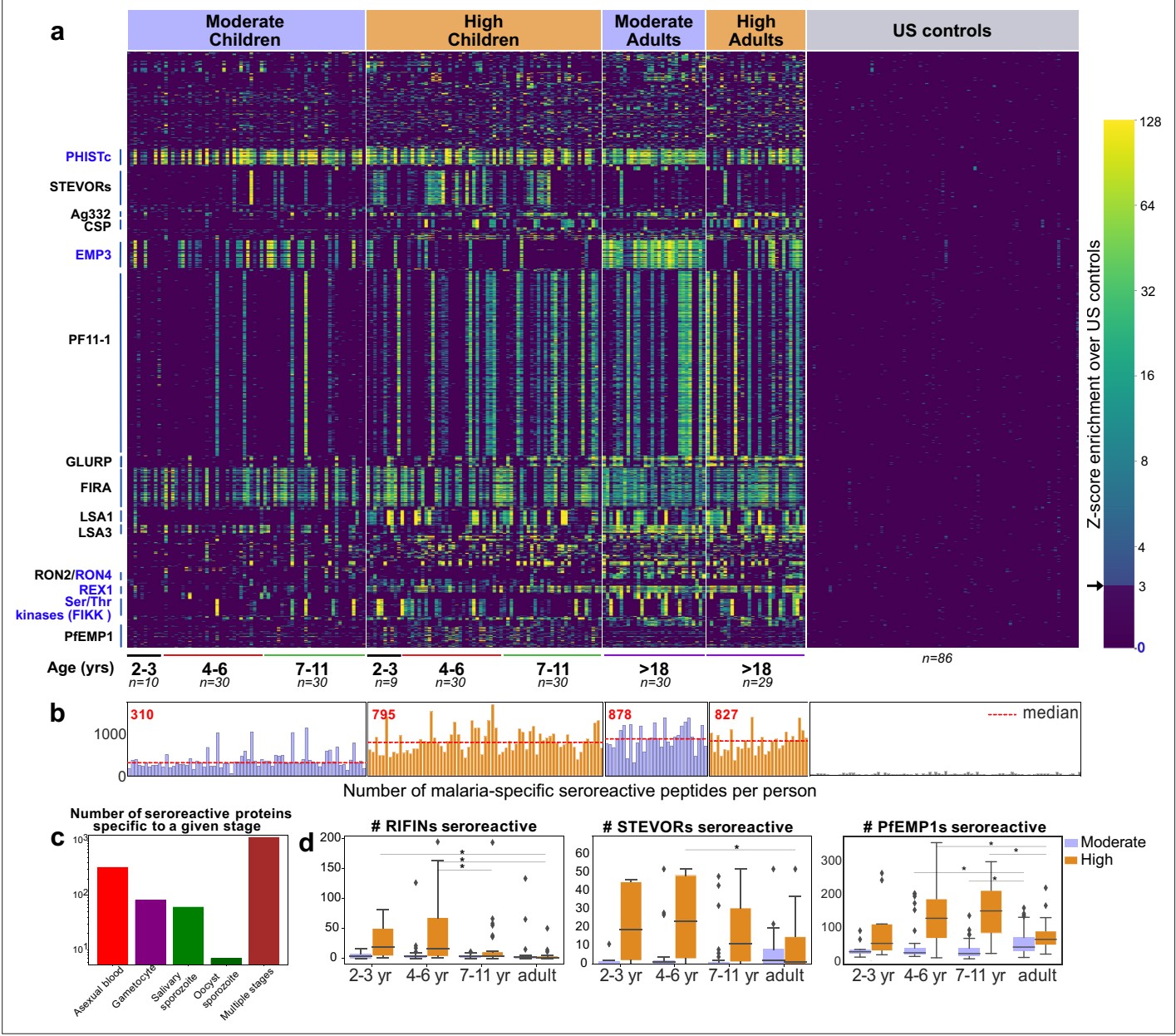

**Figure 2.** PhIP-seq with Falciparome captures known, novel antigens and relationships between age, exposure and breadth of seroreactive regions. (**a**) Heatmap of Z-score enrichment over US controls for seroreactive peptides (rows) with >10% seropositivity across different age groups in the moderate and high exposure cohorts. Peptides are sorted by protein name and samples(columns) are ordered by increasing age in each group. Examples of well-characterized (black labels) as well as under-characterized/novel (blue labels) antigens in *Plasmodium falciparum* identified with this approach are indicated. (**b**) Breadth of antibody reactivity, shown as number of seroreactive peptides in each person. Dotted red line and red text indicate median breadth for each population group. Children from the moderate transmission setting had significantly lower breadth than children from the high transmission setting as well as all adults (KS test p-value <0.05). (**c**) Number of proteins identified as seroreactive in this study that are specific to different stages. Stage classification is based on proteomic datasets in PlasmoDB (spectral count ≥ 1 for at least 1 peptide in a protein in a given stage is counted as expression) and shows enrichment of proteins from all life stages of *Plasmodium falciparum* in the human host. (**d**) Breadth of VSA reactivity, shown as number of variant proteins of RIFINs, STEVORs, and PfEMP1s seroreactive per person. In the moderate transmission setting, children had a significantly lower breadth than adults for PfEMP1 and both age groups poorly recognized RIFINs and STEVORs. In contrast, in the high transmission setting, children had a significantly (* KS test <0.05) higher breadth than adults for all three families.

The online version of this article includes the following source data and figure supplement(s) for figure 2:

**Source data 1.** GO analysis of top seroreactive proteins.

**Figure supplement 1.** Histogram of read counts of Falciparome phage library.

*Figure 2 continued on next page*

*Figure 2 continued*

**Figure supplement 2.** Technical replicates are well correlated.

**Figure supplement 3.** Target peptides are enriched in a sample-specific manner.

**Figure supplement 4.** Moving threshold analysis to determine optimal thresholds for calling peptides as seroreactive based on minimum Z-score and enrichment in a minimum number of samples in a group.

**Figure supplement 5.** Breadth of non-redundant seroreactive peptide groups per person across age and exposure.

**Figure supplement 6.** Breadth of seroreactivity in the variable regions of RIFIN and PfEMP1.

The proteins identified here overlapped substantially with antigenic proteins identified in previous protein array screens (28%, 49%, and 44% of those reported in *Camponovo et al., 2020*; *Crompton et al., 2010*; *Helb et al., 2015* respectively). However, this whole-proteome approach also identified 952 proteins not identified in the above studies. Antigens identified in previous studies may have not been enriched here because of a known limitation of PhIP-seq – it detects predominantly linear epitopes, as opposed to conformational epitopes, which would account for loss of sensitivity with respect to particular proteins. In addition, prior studies were performed in different populations and may include false positives, for example by not using a large number of unexposed human samples to account for non-specific cross-reactivity. Furthermore, in vitro protein production for arrays may not guarantee full length, or correct folding.

## Expected and new relationships between age, exposure, and breadth of seroreactive regions captured at high resolution by Falciparome

Since our cohort was stratified by age and exposure, we next set out to investigate how the overall breadth of seroreactive regions varied with age and exposure. Breadth was evaluated in two ways – (i) the total number of seroreactive peptides per person (ii) the number of non-redundant seroreactive peptide groups in each person. The latter was calculated to minimize redundant counting of potential shared linear epitopes between seroreactive peptides due to the tiled nature of the library as well as common sequences across peptides (Materials and methods). Breadth increased with age in both settings, occurring more rapidly in the higher transmission setting such that children reached a similar breadth as adults by age 11 (*Figure 2b*, *Figure 2—figure supplement 5*). As a result, children in the higher transmission setting had a significantly higher breadth than children in the moderate transmission setting. In contrast, adults in both settings had comparable breadth. Overall, these results are consistent with expected expansion of the repertoire of antibody targets with recurrent exposure to *P. falciparum* (*Crompton et al., 2010*; *Helb et al., 2015*).

Variant surface antigen (VSA) families are highly diverse, multi-member gene families in *P. falciparum* that are expressed on the surface of host erythrocytes and facilitate important functions of the parasite (*Niang et al., 2014*; *Reeder and Brown, 1996*; *Saito et al., 2017*; *Tan et al., 2015*; *Xie et al., 2021*). Expression of these genes is typically limited to one or few members of each family per parasite, presumably to evade the host immune system. Multiple variants from three VSA families were represented in the library - PfEMP1s (431 members from seven strains), RIFINs (157 3D7+118 IT), and STEVORs (32 3D7+32 IT variants), and the breadth of seroreactive variants was investigated across age and exposure based on the number of variant proteins in each family to which the VSA seroreactive peptides belonged in each person (Materials and methods; *Figure 2d*, *Figure 2—figure supplement 6*). In the moderate transmission setting, adults had a significantly higher breadth of PfEMP1 variants recognized than children (fold increase in median breadth in adults over 4–6 and 7–11 year-old children: 1.69; KS-test p-value <0.05), suggesting an age and/or cumulative exposure-dependent increase in PfEMP1 breadth in this setting, as previously observed in *Cham et al., 2009*. We note that the majority of seroreactivity in the moderate setting was present in the conserved ATS domain of PfEMP1, as opposed to the variable domains (*Figure 2—figure supplement 6*). Because sequences of variable domains may differ between the PhIPseq library and the parasites to which cohort members have been exposed, this may result in reduced sensitivity for these domains. On the other hand, both children and adults in this setting poorly recognized RIFINs and STEVORs.

In contrast, in the high transmission setting, children had a significantly higher breadth of variants recognized than adults for all three VSAs. Children of 2–6 years had the broadest responses to RIFINs (including in the variable V2 region) and STEVORS (fold increase in median breadth in 2- to 6-year-old

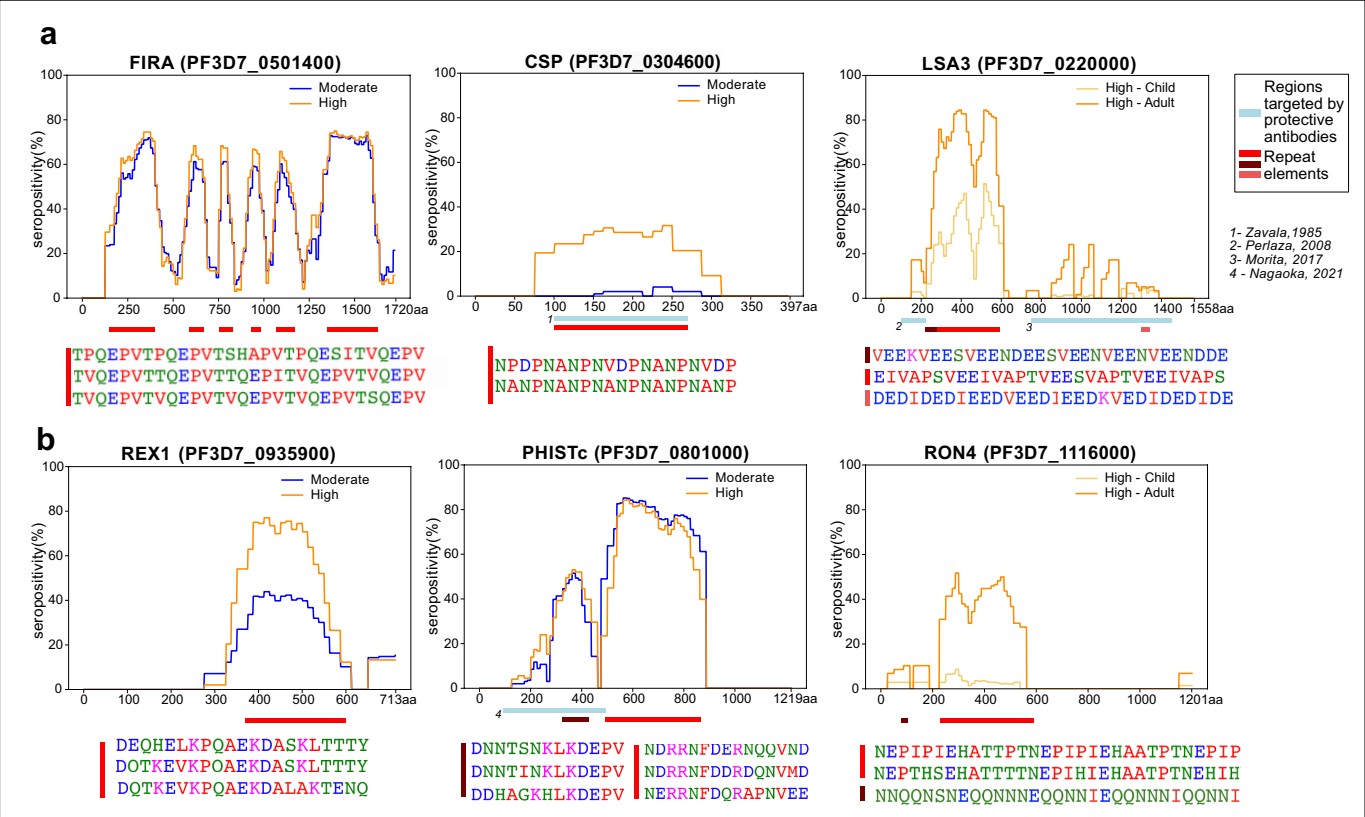

**Figure 3.** Tiled design of library facilitates high resolution characterization of seroreactive regions. (**a**) Examples of previously well-characterized antigens and (**b**) novel/previously under-characterized antigens identified in this dataset. Average percentage of people seropositive at each residue (seropositivity) based on signal from peptides spanning it are shown for each protein for different groups in the cohort. The magnitude of exposure- and age-related differences in seropositivity varies by individual protein and even within different regions of specific proteins. Reddish bars underneath each protein represent repeat elements and blue bars represent examples of regions encompassing targets of protective antibodies described in previous studies. Snapshots of sequences of repeat elements present in a protein are represented beneath the protein.

The online version of this article includes the following figure supplement(s) for figure 3:

**Figure supplement 1.** Comparison of high-resolution localization of seroreactive regions identified in this study with regions identified through a peptide-array approach.

children over adults for RIFINs: 17; KS-test p-value <0.05, for STEVORs: 21; KS-test p-value <0.05) and children of 4–11 years to PfEMP1s (including in the variable DBL domains; fold increase in median breadth in 4- to 11-year-old children over adults for PfEMP1: 2.13; KS-test p-value <0.05), suggesting a decline in responses to variants as children develop into adults in this setting. This is consistent with observations from a previous study investigating antibody responses to PfEMP1 DBLα domains in Papua New Guinea (*Barry et al., 2011*). The loss of VSA breadth in adults in the high transmission setting could be due to various reasons, including a decline in antibody levels to VSA variants due to reduced antigenic exposure, as adults have a lower parasitic load than children in this setting, or a shift in the focus of the immune response to less variable targets. It may also be possible that the decrease in VSA breadth in adults reflects a transition from recognition of linear epitopes to conformational epitopes, which may not be detected in this assay (or prior microarray assays).

## Tiled design of library facilitates high-resolution characterization of seroreactive proteins

The short length and tiling design of the peptides in this library facilitated high-resolution characterization of antigenic regions within seroreactive proteins. Representative examples from previously characterized proteins, such as Falciparum Interspersed Repeat Antigen (FIRA), Circumsporozoite Protein (CSP) and Liver Stage Antigen (LSA3) are shown (*Figure 3a*), where known epitopes consisted

of short amino-acid motifs repeated multiple times within the proteins ('repeat elements'). Comparison with a previous study using a high-density linear peptide array covering a subset of antigens showed substantial overlap of the seroreactive regions within these antigens (*Jaenisch et al., 2019*; *Figure 3—figure supplement 1*), although some differences were apparent. Differences in the length of peptides as well as nature of display (linear 15-aa peptides on an array versus phage display of 62-aa peptides) are potential explanations for these discrepancies.

Importantly, high-resolution maps of seroreactivity for over 1000 proteins were characterized for the first time in our dataset (*Figure 3b*). A notable example is, PHISTc (PF3D7_0801000), which has previously been described as an antigenic protein, but not dissected at high resolution (*Baum et al., 2013*; *Dent et al., 2015*). It is exported from the parasite during the asexual blood stage and has unknown function, although mildly protective antibodies have been described against the N-terminal segment (*Nagaoka et al., 2021*). Another example is RON4 (Rhoptry Neck Protein 4), part of the moving junction during merozoite invasion of the host (*Morahan et al., 2009*) and is also critical for sporozoite invasion of hepatocytes (*Giovannini et al., 2011*).

Beyond the overall breadth of seroreactive peptides, the dataset facilitated a high-resolution lens for investigating the effect of age and exposure on seroreactivity to individual proteins. For instance, as expected (*Kazmin et al., 2017*), we observed exposure-dependent seropositivity at the B-cell epitope in CSP targeted by the RTS,S vaccine (NANP repeating sequence; *Figure 3a*). The magnitude of exposure- and age-related differences in proportion seropositive varied by individual protein and even within different regions of specific proteins (*Figure 3*, *Supplementary file 4*), highlighting the importance of dissecting responses to different antigenic regions within seroreactive proteins.

## Seroreactive proteins contain more repeat elements than non-seroreactive proteins

A prominent feature that stood out following high-resolution characterization of seroreactive regions was the presence of repeat elements, where identical or similar motifs were repeated in tandem or with gaps within a given protein (*Figure 3*). Previous studies focused on individual or targeted subsets of antigens in *P. falciparum* have highlighted the immunogenic nature of short amino acid repeat sequences (*Davies et al., 2017*). The proteome of *P. falciparum* is notoriously rich in such sequences; however, their functions have remained enigmatic and their properties have been difficult to characterize. To systematically investigate the association of seroreactivity with these elements, repeats throughout all coding sequences were first identified using RADAR (Rapid Automatic Detection and Alignment of Repeats) (*Madeira et al., 2019*) and then compared to both PhIP-seq seroreactive and non-seroreactive proteins. The number of repeats in each protein sequence was significantly higher in the seroreactive proteins in comparison to non-seroreactive proteins (median number of repeats per protein: seroreactive proteins – 20; non-seroreactive proteins – 6; p-value = <0.001, based on 1000 iterations [1636 proteins per iteration] of random sampling of the non-seroreactive set; *Figure 4a*).

## Seroreactive peptides, within seroreactive proteins, contain more repeat elements than non-seroreactive peptides

Next, we investigated if seroreactive regions within seroreactive proteins were enriched for repeat elements. Because the Falciparome is composed of overlapping peptides tiled across each gene, the contribution of individual peptide sequences within each seroreactive protein can be further classified into those that are seroreactive vs. those from the same protein that are non-seroreactive. This enables a comparison of repeat elements among seroreactive and non-seroreactive peptides within each protein sequence.

To accomplish this, a k-mer approach was used to characterize repeat elements (*Figure 4b*, Methods). Briefly, the frequency of all biochemically similar k-mers of sizes 6-9aa (approximately the size of a linear B-cell epitope) was calculated for each protein. Then, each peptide in the protein was assigned a repeat index based on the maximum intra-protein frequency of any repeat element it encompassed. To minimize redundant representation, multiple peptides from a given protein deriving their repeat indices from the same repeat element were collapsed such that a repeat element was represented only once for each protein (*Figure 4b*). In this manner, the set of all 5171 non-VSA seroreactive-peptides was collapsed based on their repeat elements to a set of 3091 non-redundant

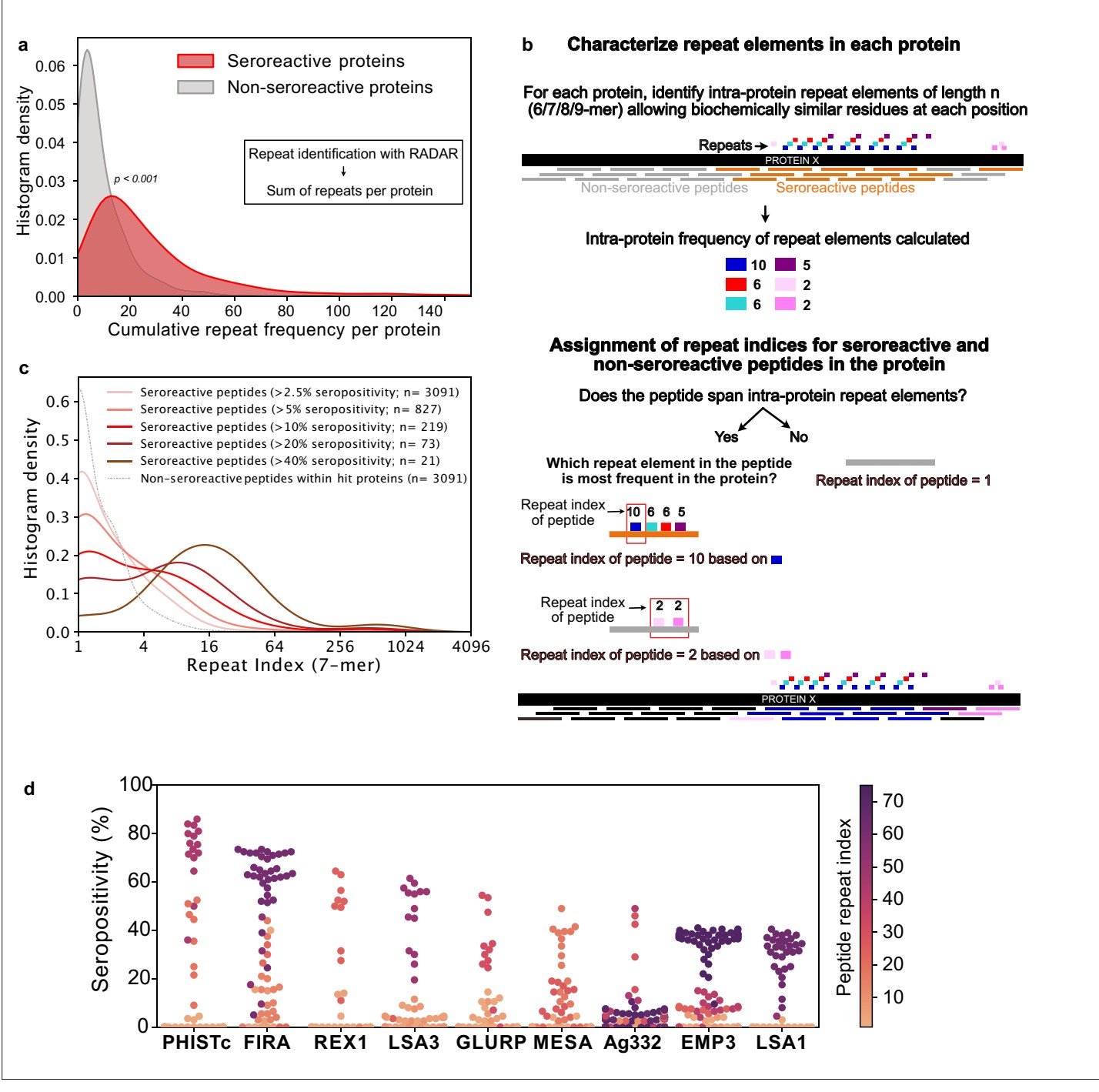

**Figure 4.** Repeat elements are more enriched in seroreactive peptides within seroreactive proteins than non-seroreactive peptides. (**a**) Distribution of cumulative frequency of repeat elements per protein is significantly higher (KS test p-value <0.05) in the seroreactive protein set than a randomly sampled subset of non-seroreactive proteins (1000 iterations). (**b**) Pipeline to compute the representation of repeats in each peptide as repeat index. (**c**) Distribution of repeat indices is significantly higher (KS test p-value <0.05) in seroreactive peptides than a randomly sampled subset of non-seroreactive peptides within seroreactive proteins (1000 iterations). Distribution of repeat indices also significantly increases with increase in seropositivity (KS test p-value <0.05 between all successive distributions). (**d**) Seropositivity of all peptides (dots) colored by their repeat indices in the top 9 most seropositive repeat-containing proteins shows enrichment of repeat elements in peptides with high seropositivity.

The online version of this article includes the following figure supplement(s) for figure 4:

**Figure supplement 1.** Distribution of repeat indices of seroreactive and non-seroreactive peptides within hit proteins for different lengths and degeneracy of the repeating motif.

seroreactive peptides. The non-seroreactive peptides within each seroreactive protein were also collapsed similarly.

Overall, seroreactive peptides yielded significantly higher repeat indices than non-seroreactive peptides from seroreactive proteins, and this trend was more pronounced as a function of seropositivity (*Figure 4c*). The median repeat index for non-seroreactive peptides was 1, while the median index for >10% and >40% seropositivity was 3 and 13 respectively, for a kmer of size 7 (KS test p-value <0.05 between successive distributions). These results suggest that seroreactive peptides are dominated by repeat elements and those with higher seropositivity also have progressively higher repeat indices. Examination of individual proteins, including well characterized repeat-containing antigens such as FIRA, LSA1, LSA3, MESA, and GLURP, illustrate the relationship between seropositivity and repeat index (*Figure 4d*). This relationship was consistently observed, regardless of kmer size from 6 to 9aa, and was insensitive to the level of degeneracy or biochemical similarity used for determining repeat matches (*Figure 4—figure supplement 1*). However, the presence of a repeat element within any given peptide does not necessarily imply that the peptide will be seroreactive.

Taken together, these data indicate that seroreactive proteins tend to be repeat-containing proteins, and within these proteins, the individual seroreactive peptides tend to be those that contain the repeats. Furthermore, seroreactive regions that are shared widely among individuals tend to feature higher numbers of repeat elements.

## Seropositivity is more exposure-dependent and short-lived in children for peptides containing repeat elements than those without repeats

To investigate whether the breadth of seroreactive repeat-containing peptides differed depending on exposure-setting and age, seroreactive peptides were first binned into two categories: those with repeats, and those without. Specifically, seroreactive peptides with a 7-mer repeat index ≥ 3 were binned together as 'repeat-containing peptides' and those with a 7-mer repeat index ≤ 2 were binned as 'non-repeat peptides'. For the set of non-repeat containing peptides, breadth (number of non-repeat peptides enriched per person) was significantly higher in adults than children in both exposure settings (percent increase in median breadth in adults over 4- to 6-year old children: moderate setting – 28%; high setting – 20%; *Figure 5a*). However, within each set of age groups, there was no significant difference in breadth between the two exposure settings.

For repeat-containing seroreactive peptides, breadth was calculated as follows. Each repeat-containing seroreactive peptide was defined by the 7-mer (repeat element) that was used to calculate its repeat index as described above. To avoid redundant counting, all repeat-containing peptides from a given protein defined by the same repeat element were collapsed and counted only once. Similar to non-repeat peptides, breadth of these peptides was higher in adults than children, reaching a similar level in both exposure settings (percent increase in median breadth in adults over 4- to 6-year-old children: moderate setting – 193%; high setting – 56%). In contrast to non-repeat peptides however, there was an exposure dependence in the responses to repeat-containing peptides with age, such that children living in the high versus moderate exposure setting had twice the breadth of repeat-containing peptides, reaching the same level in adulthood in both settings (*Figure 5b*). These results were consistently observed with different thresholds for categorizing repeat-containing peptides (repeat index ≥ 4 or 5; *Figure 5—figure supplement 1*). Investigation of individual repeat elements recapitulated this trend and showed higher seropositivity in the high exposure setting compared to moderate exposure in children, but not adults (*Figure 5—figure supplement 2*). There were a small number of notable exceptions, including repeat elements from PHISTc (PF3D7_0801000), LSA3 (PF3D7_0220000), and FIRA (PF3D7_0501400), none of which showed a transmission setting-dependent response in children (*Supplementary file 5*).

Samples from the two exposure settings differed not just by exposure, but also with respect to time since most recent infection, reflecting the differing epidemiology of infection in these settings. In the moderate exposure setting, the median number of days since last infection was 100, whereas over 65% of the samples from the high exposure setting were taken during periods of active infection. The difference in seroreactivity to repeat-containing peptides observed here between the settings could therefore emerge from two related mechanisms. In the first, the difference could be driven by a requirement for a minimum level of cumulative exposure to the target repeats to generate a robust response. In the second, the antibody response to repeats may be inherently less durable, leading to

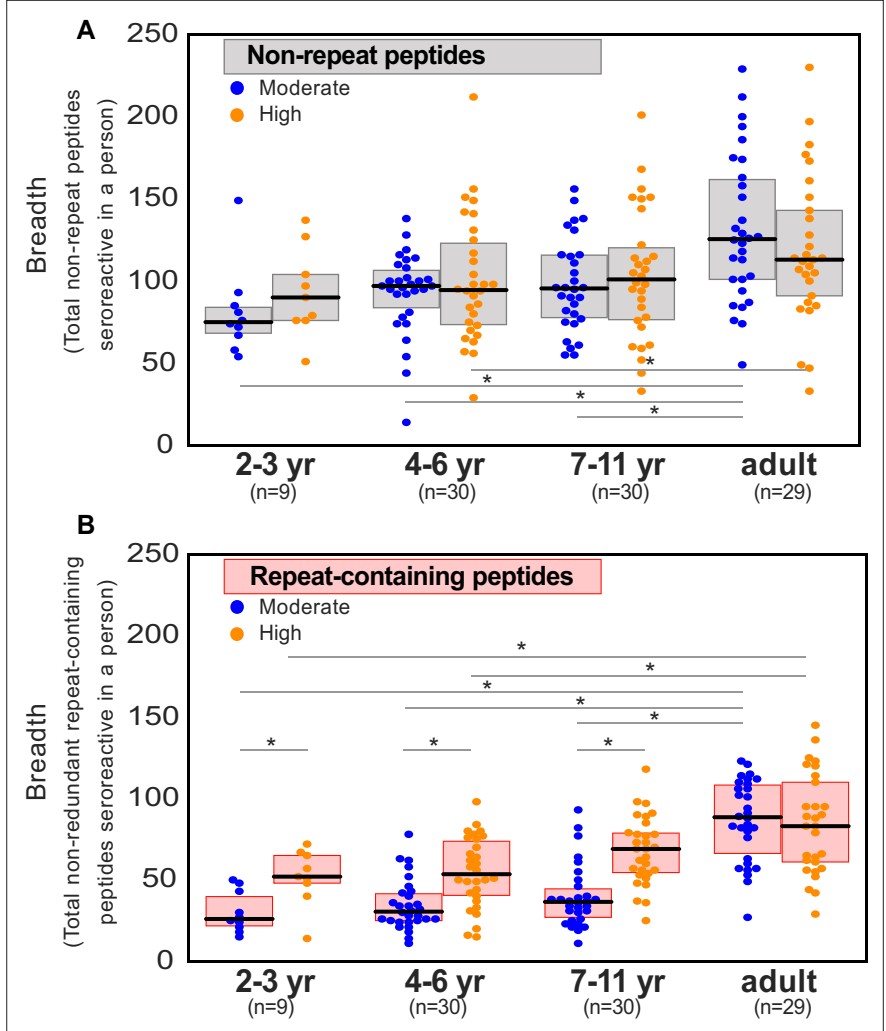

**Figure 5.** Breadth of seroreactive repeat-containing peptides, but not non-repeat peptides, increases with exposure in children. (**a**) Breadth of seroreactive non-repeat peptides per person is not significantly different (KS-test p-value >0.05) between the two exposure settings within each age group. (**b**) Breadth of seroreactive repeat-containing peptides per person is significantly higher (KS-test p-value <0.05) in the high exposure setting than in the moderate exposure setting within the three groups in children, but not adults.

The online version of this article includes the following figure supplement(s) for figure 5:

**Figure supplement 1.** Breadth of repeat-containing peptides per person using different repeat index thresholds for categorizing repeat-containing peptides.

**Figure supplement 2.** Seropositivity of individual seroreactive repeat elements increases with exposure in children, but not adults.

**Figure supplement 3.** Controlling for time since infection status, breadth of seroreactive repeat-containing peptides, but not non-repeat peptides, still shows an increase with exposure in children.

**Figure supplement 4.** Breadth of seroreactive repeat-containing peptides, but not non-repeat peptides, wanes with increased time since infection in the moderate exposure setting in children.

rapid waning in the absence of frequent exposure. Each of these two possibilities were investigated below.

First, to determine the effect of exposure while controlling for infection status, children (ages 2–11 years) were combined to afford sufficient statistical power and were classified as actively infected or 60–120 days from infection (those infected 1–59 days prior were excluded from this analysis to look at a time point well past infection). For repeat containing peptides, breadth was significantly increased in the high exposure setting relative to moderate exposure, regardless of infection status at the time

of sampling. This difference was not observed for non-repeat containing peptides (*Figure 5—figure supplement 3*). These data suggest a dependency on cumulative exposure for the observed differential antibody responses in children to repeat-containing peptides relative to non-repeat peptides.

Next, durability of responses was investigated by characterizing differences in breadth with, time since last infection, in each exposure setting (*Figure 5—figure supplement 4*). For repeat containing peptides, a significant decrease in breadth was evident between the most recent infections (0–30 days) and the longest (240–360 days) in the moderate exposure setting, while the difference in the high exposure setting between 0–30 days and 60–120 days was not significant; notably everyone in this setting was infected within the prior 120 days. No significant difference was evident for non-repeat peptides in either setting. Together, these results are consistent with the notion that the antibody response to repeat containing peptides are short lived relative to non-repeat peptides. Overall, the above data show that antibody responses to repeat-containing peptides may be less efficiently acquired and/or maintained in children than non-repeat peptides, but plateau at the same level of prevalence in adulthood.

## Extensive sharing of motifs observed between seroreactive proteins, particularly the PfEMP1 family

While repeat elements within individual proteins were explored in the previous section, similar or identical motifs may also be shared among different proteins. If these motifs are a part of an epitope, then antibodies and BCR specific to a motif can potentially cross-react with the motif variants in different proteins, depending on accessibility and other factors. Identifying such shared motifs serves as the first step in exploring potential cross-reactivity between individual seroreactive proteins, and to identify them, a systematic investigation was performed.

First, enriched kmers (6–9 amino acids) were identified by collecting those present in a significantly (FDR-adjusted p-value <0.001) higher number of seroreactive peptides (9927) than a random sampling (1000 iterations) of 9927 peptides from the whole library. From this collection, enriched kmers that were shared by different seroreactive proteins were identified as 'inter-protein motifs' (*Figure 6a*). Using a kmer size of 7, and allowing for up to two biochemically conservative substitutions, a total of 911 significantly enriched inter-protein motifs were identified, representing 509 seroreactive proteins (*Supplementary file 6*). Limiting the selection of inter-protein motifs to only the most significantly enriched motif per seroreactive peptide (the motif with the lowest p-value among all motifs in each peptide) yielded 417 significant inter-protein motifs, from a similar number of proteins (507). As expected, increasing the kmer size, or further constraining the number of allowed substitutions resulted in fewer identified motifs (*Supplementary file 7*). For the subsequent analysis, we show results with a kmer size of 7, which is in the range of average length of a linear B-cell epitope (*Buus et al., 2012*). As expected, previously described cross-reactive epitopes between antigens were well represented, such as the glutamate-rich motifs in Pf11-1 and Ag332 (*Mattei et al., 1989*), among others (*Figure 6a*). Taken as a group, the collected motifs had a lower hydrophobicity index (mean Kyle-Doolittle=−1.95), a lower net charge (mean = −0.47; at pH 7), enrichment of charged glutamate, lysine, asparagine, and aspartate residues and depletion of cysteine and hydrophobic residues than a random set of motifs in the proteome (*Figure 6—figure supplement 1*). These biochemical characteristics are consistent with those observed in prior studies of residues in B cell epitopes (*Akbar et al., 2021*; *Rubinstein et al., 2008*).

The design of the programmable phage display library used here features 62 amino acid peptides tiled with a 25 amino acid step size, yielding an overlap of 37 amino acids for sequential fragments, and 12 amino acids for every second fragment (*Figure 6—figure supplement 2a*). The design provides for localization of seroreactive sequences to the region of overlap when considering adjacent fragments. For all except the first and last two peptides in each protein (85% of peptides in the library), the seroreactive region can theoretically be narrowed down to a 12-13aa segment within the peptide. Given that B cell linear epitopes are typically 5–12 amino acids in length (*Buus et al., 2012*), the 12-13aa mapping provides a near-epitope resolution.

To test the notion that the inter-protein motifs within each peptide are actually the elements associated with the observed seroreactivity, we leveraged the tiled peptide library design by comparing inter-protein motif carrying peptides with overlapping and adjacent peptides (*Figure 6—figure supplement 2b*). The maximum seropositivity among peptides containing an inter-protein motif

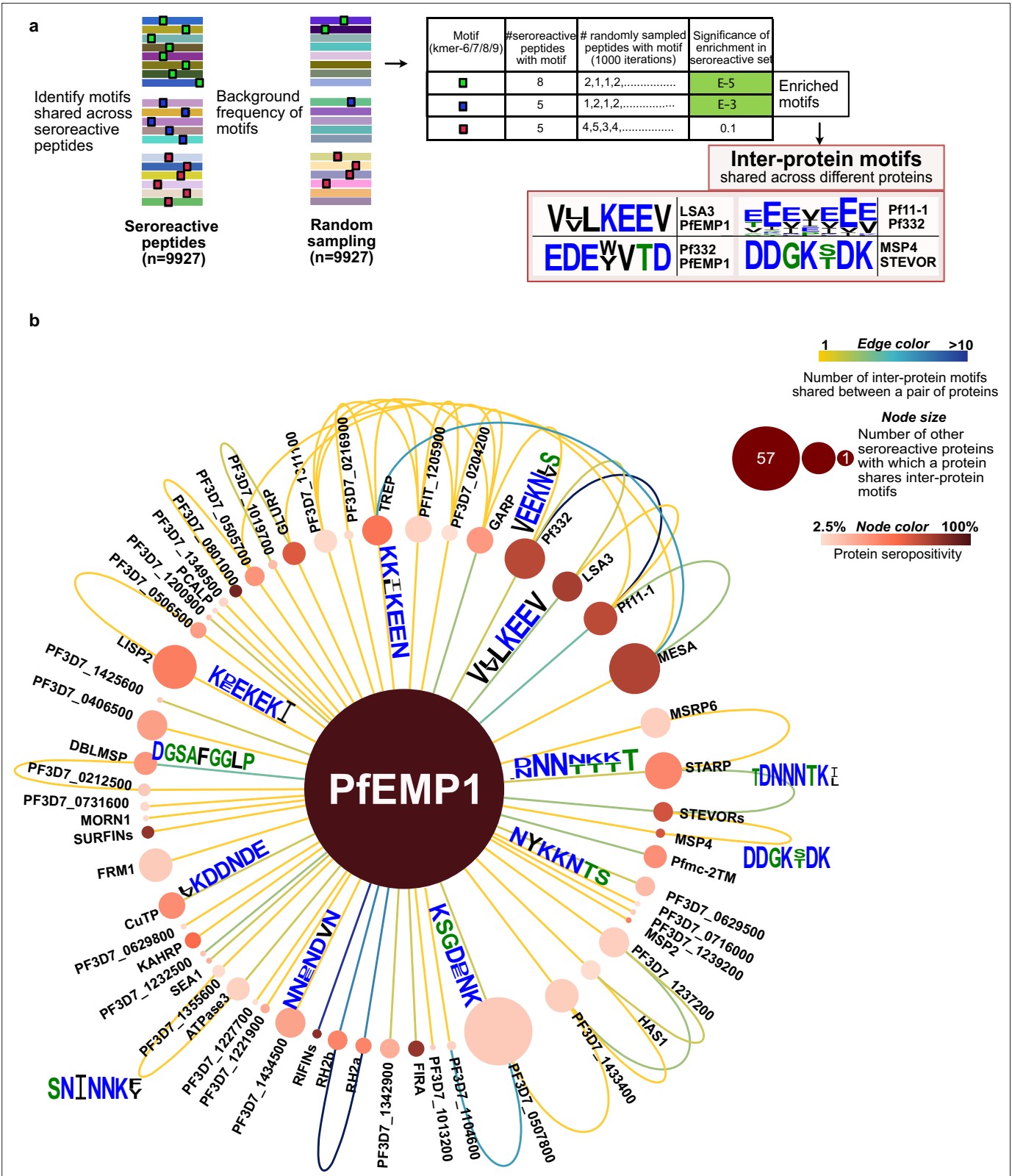

**Figure 6.** Extensive sharing of motifs observed among seroreactive proteins, with the most shared with PfEMP1 family. (**a**) Pipeline to identify inter-protein motifs (6-9aa) significantly enriched (FDR <0.001) in seroreactive peptides from different seroreactive proteins (different colors) over background. Background for each motif was estimated based on the number of random peptides possessing the motif in 1000 random samplings of 9927 peptides. Examples of inter-protein motifs and seroreactive proteins sharing them are also shown. (**b**) Network of PfEMP1 sharing inter-protein motifs with other

*Figure 6 continued on next page*

*Figure 6 continued*

seroreactive proteins based on 7-aa motifs with up to two conservative substitutions. PfEMP1 shared inter-protein motifs with the greatest number of other seroreactive proteins.

The online version of this article includes the following figure supplement(s) for figure 6:

**Figure supplement 1.** Biochemical characteristics of inter-protein motifs.

**Figure supplement 2.** Inter-protein motifs are associated with seroreactivity.

**Figure supplement 3.** Co-occurrence of reactivity to peptides containing inter-protein motifs from different proteins within individuals.

**Figure supplement 4.** Histogram of number of other seroreactive proteins with which a seroreactive protein shares inter-protein motifs.

**Figure supplement 5.** Network of seroreactive proteins outside the PfEMP1 network.

was on average 54-fold higher than the maximum seropositivity among overlapping peptides not containing the motif (using a pseudo-seropositivity of 0.1% for peptides with 0% seropositivity to facilitate fold change calculation), suggesting a strong association between seroreactivity and the inter-protein motif itself, not just the whole peptide within which it resides (comparison of median seropositivity yielded a similar result). Furthermore, a similar result was observed when the same analysis was done with all the significantly enriched kmers (*Figure 6—figure supplement 2c*).

Then, to evaluate potential cross-reactivity, we measured the co-occurrence of reactivity to peptides containing inter-protein motifs from different proteins within the same individual. A cumulative profile was then created from the aggregation of the results across all individuals in the study, and then compared to a background distribution drawn by random sampling of peptides without inter-protein motifs (*Figure 6—figure supplement 3*). We observed that shared seroreactivity of inter-protein motifs within individuals was significantly higher than in the randomly drawn peptides lacking these motifs (KS test p-value <0.01). Overall, these data suggest a possibility for cross-reactivity to these motifs within individuals. However, it is important to note that these data were derived from complex polyclonal responses in each person, and do not represent direct evaluation of the cross-reactivity of single antibodies.

On average, each inter-protein motif was shared by 3 seroreactive proteins (median = 2). Among the 509 seroreactive proteins, each of them shared inter-protein motifs with six other proteins on average (median = 3), (*Supplementary file 8*, *Figure 6—figure supplement 4*). Visualized as a network (*Figure 6b*), the PfEMP1 family of proteins formed a central hub to which a large number of other seroreactive proteins were connected. The PfEMP1 family of proteins possessed at least 90 shared inter-protein motifs, and this family shared those motifs with the greatest number of other seroreactive proteins (57) compared to all other proteins in this analysis. Approximately five times as many proteins shared connections with PfEMP1 than would be expected by chance (PfEMP1 shared motifs with 12–16 other proteins using a set of 9927 peptides consisting of PfEMP1 seroreactive peptides + random non-PfEMP1 peptides). Seroreactive proteins sharing motifs with PfEMP1 included many of the proteins with the highest measured seropositivity, such as RIFINs, SURFINs, FIRA, and PHISTc. This extent of sharing was driven, in part, by the number of PfEMP1 sequences included in the analysis. This was apparent when the same analysis performed with a reduced diversity of PfEMP1 sequences in the seroreactive peptide set (using PfEMP1 peptides from only PF3D7 and PFIT genomes instead of 7 genomes) resulted in PfEMP1 sharing motifs with 32 seroreactive proteins instead of 57. This suggests that the extent of sharing for PfEMP1 observed in this study may only be a small fraction of that occurring in the extensive natural diversity of PfEMP1 variants in circulating parasites.

Outside the main network driven by PfEMP1, 495 seroreactive proteins were also found to be highly connected to each other through motif sharing (*Figure 6—figure supplement 5a*). A large proportion of proteins with high seropositivity were connected (80% and 58% of proteins with >30% and 10–30% seropositivity, respectively). This included proteins like GARP, LSA3, Pf332, Pf11-1, and MESA (*Figure 6b*, *Figure 6—figure supplement 5b*). As observed for the full set of inter protein motifs, motifs shared by the subset of proteins with >30% seropositivity also consisted predominantly of charged glutamate, lysine, asparagine, and aspartate residues (*Figure 6—figure supplement 5b*). Since the analysis used here to identify inter-protein motifs allowed only up to two conservative substitutions in 7-mer motifs, the similarity of motifs in the network in *Figure 6—figure supplement 5b* suggests that with a less stringent threshold of identifying motifs, these proteins would be even more

highly connected. Moreover, 80% of proteins in this network had reported expression in the asexual blood stage of the lifecycle of *P. falciparum* (PlasmoDB), suggesting temporally concordant presence of proteins sharing motifs within their seroreactive regions.

These results indicate that the interprotein motifs are strongly associated with seroreactivity and are extensively shared across seroreactive proteins, including among regions highly targeted by the antibodies. Furthermore, PfEMP1 shares motifs with the greatest number of other seroreactive proteins, partly driven by the sequence diversity of PfEMP1 variants.

## Discussion

Using a customized programmable phage display (PhIP-seq) library, we have evaluated the proteome-wide antigenic landscape of the malaria parasite *P. falciparum*, using the sera of 198 individuals living in two distinct malaria endemic areas. This approach readily identified previously known antigens, including proteins that are targets of protective antibodies, as well as novel antigens. In our study, we characterized features of *P. falciparum* antigens that could potentially contribute to the inefficient acquisition and maintenance of immunity to malaria. Repeat elements were found to be commonly targeted by antibodies, and had patterns of seropositivity that were less durable and more dependent on exposure than non-repeat peptides. Furthermore, extensive sharing of motifs associated with seroreactivity was observed among hundreds of parasite proteins, indicating potential for extensive cross-reactivity among antigens in *P. falciparum*. These data suggest that repeat elements– a common feature of the *P. falciparum* proteome, and shared motifs between antigenic proteins could have important roles in shaping the nature and development of the immune response to malaria.

To map the antigenic landscape, PhIP-seq for *P. falciparum* offers several attractive advantages. The library described here contains >99.5% of the proteome, including variants for several antigenic families, surpassing the coverage of other existing proteome-wide tools for *P. falciparum* (**Camponovo et al., 2020**; **Morita et al., 2017**), while simultaneously providing high-resolution characterization of antigens (up to 12–13 aa regions within peptides). Unlike peptide arrays, the platform converts a proteomic assay into a genomic assay, leveraging the massive scale and low-cost nature of next-generation short-read sequencing. The result is a cost-effective and scalable system, allowing for the processing of hundreds of samples in parallel. Finally, an important aspect of all phage display systems is the ability to sequentially enrich, release phage, and repeat, thus greatly amplifying the signal to noise (**O'Donovan et al., 2020**; **Smith and Petrenko, 1997**). Only one published study to date evaluates responses to more than a quarter of the proteome (**Camponovo et al., 2020**), inherently limiting the scope of potential targets interrogated. Twenty-eight percent of the hits described in this study of individuals living in Tanzania and exposed to various doses of PfSpz vaccine overlapped with the hit proteins described in our study (**Camponovo et al., 2020**). The limited overlap may be due to multiple factors, including differences in the characteristics of individuals sampled, the use of vaccine, and determination of seropositivity based on technical as opposed to biological controls (sera from unexposed individuals).

The near-epitope resolution provided by this platform allowed a systematic investigation of targets of antibodies. Targets with high seropositivity were observed to be significantly enriched for repeat elements. In some previous reports, the elevated antigenic potential of repeat elements has been noted (**Davies et al., 2017**); however, the proteome-wide approach described here demonstrates that a large collection of proteins containing these elements are highly immunogenic. The high immunogenicity of repeat sequences observed here may be the result of competitive advantages that B cell clones could encounter when binding to higher valency epitopes, as opposed to single copy epitopes. Evidence from experimental inoculations of antigens with differing repeat numbers support this notion (**Kato et al., 2020**). Moreover, tandem repeat regions are predicted to be intrinsically disordered, which in turn have favorable predictions as linear B cell epitopes (**Guy et al., 2015**). Notably, this high immunogenicity can potentially restrict responses to other epitopes within the antigen, as has been reported for responses to protective non-repeat epitopes in the circumsporozoite protein (CSP; **Chatterjee et al., 2021**).

A key finding of this study is the difference in seroreactivity to repeat-containing peptides in children vs. other peptides, with the breadth of seroreactivity to repeat-containing peptides increasing more quickly with age in the high versus moderate exposure setting and decreasing with increase in time since infection. These results suggest that antibody responses to repeats are more likely to be

exposure-dependent and/or short-lived in children than responses to non-repeats. There were a few exceptions, including repeats from FIRA, PHISTc, and LSA3, that did not show an exposure-setting dependent difference in seropositivity, suggesting that factors beyond the repeated nature of the epitope influence the nature of the response. Regardless, the predominance of repeat containing peptides in antibody targets, along with the remarkable abundance of these peptides in the *P. falciparum* proteome, suggests a possible strategy evolved by the parasite for the purpose of diverting the humoral response towards short-lived or exposure-dependent responses.

The hypothesis of less durable antibody responses to repeat antigens in *P. falciparum* can be reconciled with a model in which repeating epitopes favor extrafollicular B cell responses, which are typically short-lived (*Cockburn and Seder, 2018*; *Schofield, 1991*). This is based on the potential of repeat epitopes in an antigen to interact with multiple BCRs on naive B-cells, thereby conferring high binding strength and sufficient activation to direct these cells into an early extrafollicular response and production of short-lived plasmablasts. Several studies provide support to this model, where strong binding of BCR to the antigen, including through increased valency, increases the production of plasmablasts (*Kato et al., 2020*; *O'Connor et al., 2006*; *Ochiai et al., 2013*; *Paus et al., 2006*; *Schwickert et al., 2011*). This could also happen via a T-cell-independent response, as has been reported for some repeat antigenic structures (*Schofield, 1991*; *Schofield and Uadia, 1990*). On the other hand, the effect on germinal centers (GC), which result in long-lived plasma cells (LLPCs) and isotype-switched memory cells, is unclear. While some studies have reported no change or a decrease in the formation of GCs (*O'Connor et al., 2006*; *Ochiai et al., 2013*; *Paus et al., 2006*) with increased strength of interaction between antigen and B-cells, some have reported an increase (*Kato et al., 2020*; *Schwickert et al., 2011*), although it is unclear whether the latter were productive GCs. More insights come from a few studies that measured the outcome of GCs following increased strength of BCR-antigen interaction and these have reported a decrease in LLPCs (*Fink et al., 2007*) and IgG-switched memory cells (*Pape et al., 2018*; *Taylor et al., 2015*). If repeat antigens in *P. falciparum* follow this pattern, an expected outcome would be defective formation of LLPCs and memory B cells.

Another major finding of this study is the extensive presence of inter-protein motifs among seroreactive proteins. Since a strong association with seroreactivity was observed for these motifs and there was evidence of shared reactivity among peptides containing the motif from different proteins, they may potentially represent cross-reactive epitopes. Definitive confirmation of cross-reactivity could be demonstrated with targeted studies using monoclonal antibodies. Furthermore, whether these inter-protein motifs are cross-reactive in vivo is unclear and may depend on expression timing and accessibility to the immune system, among other factors.

Analogously, seroreactive repeat elements with non-identical repeating units could represent cross-reactive epitopes within proteins. Extensive presence of potential cross-reactive epitopes in *P. falciparum* antigens may play an important role in influencing the quality of the immune response to malaria. While it could be advantageous for the host if multiple parasite proteins could be targeted by antibodies through cross-reactivity, simultaneous presence of cross-reactive epitopes could alternatively frustrate the affinity maturation process due to conflicting selection forces, as was observed for variant HIV antigens (*Wang et al., 2015*). Further, recurrent exposure may be necessary for the generation of strong cross-binding antibodies to cross-reactive epitopes (*Murugan et al., 2020*). Thus, the extensive presence of cross-reactive epitopes, both within and between antigenic proteins in *P. falciparum*, could represent an evolutionary strategy aimed at limiting high-affinity antibodies in favor of lower affinity, cross reactive antibodies. In essence, the large number of shared seroreactive sequences in *P. falciparum* may represent a complex immune counter measure, resulting in inefficient immunity acquisition which requires extensive exposure. The atlas of seroreactive repeat elements and inter-protein motifs from this study will be useful for future investigations in understanding their impact on the quality of immune response to malaria.

The PfEMP1 family shared inter-protein motifs with the greatest number of other antigens in this study. This was driven in part by the wide diversity of PfEMP1 variants, indicating that as one becomes naturally exposed to different PfEMP1 variants (*Cham et al., 2009*), there may be an increase in not only the sequence diversity, but the number of cross-reactive epitopes that the immune system encounters. Possessing cross-reactive epitopes with other antigens could result in binding of pre-existing antibodies to the new variants, which could be disadvantageous to the host if binding strength is weak. Further, cross-reactivity may inhibit generation of antibodies specific to the new variant due

to original antigenic sin (*Vatti et al., 2017*). Thus, the mechanism of evolving PfEMP1 variants within a network of shared sequences with other antigens could be another strategy evolved by the parasite for immune evasion. On the other hand, binding of new variants to pre-existing antibodies may be advantageous to the host if those antibodies are effective against the new variants.

While phage display of small peptides yields high-resolution discrimination of linear epitopes, this approach may not capture antibodies binding to conformational epitopes. Therefore, such epitopes are likely to be missed by this assay, although polyclonal responses are frequently a mixture of linear, partially linear, and conformational epitopes. Reassuringly, we observed a large-scale enrichment of *P. falciparum* peptide sequences in exposed individuals when compared with control sera from the US. This suggests that the humoral immune system of exposed individuals acquires an extensive and diverse set of *P. falciparum* targets, including thousands of linear sequences. The bias towards linear epitopes may have increased the relative detectability of repeat regions by this assay since they often form intrinsically disordered regions. However, that would not account for the observed differences between exposure settings for children and adults. Furthermore, antibodies that depend on epitopes with disulfide linkages and post-translational modifications for binding would likely not be enriched using phage-presented peptides. Another limitation of our study is that the PhIP-seq technique does not inherently provide quantitative measures of antibody affinity or titer, as many factors influence the actual number of reads recovered after immunoprecipitation, such as starting copy number in the library and non-specific interactions with beads. Instead, our analysis relies on per peptide relative (ratiometric) enrichments, using non-malaria control (n=86) sera as the basis for comparison, which also serves to remove non-specific enrichments. We imposed a stringent filter to minimize false positives by requiring that each seroreactive peptide be enriched in at least five Ugandan samples over control sera. While this excluded possible seroreactive peptides unique to a single sample, the resulting sequences that passed were those that exhibited a minimum level of sharing among multiple individuals, thereby enriching for those seroreactive peptides that represent common serological responses to malaria.

Findings from this study could have important implications on malaria vaccine design. Results from our study suggest that that in natural infections in children, repeat regions in *P. falciparum* could lead to an exposure-dependent and/or short-lived antibody response to a higher degree than for non-repeat regions. While we recognize that vaccine induced immunity is distinct from naturally acquired immunity, this potential limitation should be considered when evaluating repeat-containing antigens as vaccine targets. Further, given that highly immunogenic regions in natural immunity to malaria are predominantly repeats and there is widespread presence of potential cross-reactive epitopes across many proteins, whole-parasite vaccines may also be susceptible to similar limitations. If the findings from this study translate to vaccine-induce immune responses, non-repetitive, unique antigenic regions may be more effective targets.

# Materials and methods

## Key resources table

| Reagent type (species) or resource | Designation | Source or reference | Identifiers | Additional information |
|---|---|---|---|---|
| Strain, strain background (*E. coli*) | BLT5403 | Novagen/EMD Millipore, T7 Select Kit | Cat# 70550–3 | |
| Strain, strain background (T7 Bacteriophage) | T7 vector arms, Packaging extract | Novagen/EMD Millipore, T7 Select Kit | Cat# 70550–3 | |
| Genetic reagent (T7 Bacteriophage library) | Falciparome | Made in this study | | See Materials and Methods |
| Biological sample (Humans) | Ugandan cohort plasma | *Kamya et al., 2015*, *Rek et al., 2016*; *Yeka et al., 2015* | | |
| Biological sample (Humans) | US control plasma | New York Blood Center | | |
| Antibody | Anti-Glial Fibrillary Associated Protein (rabbit, polyclonal) | Agilent | Cat# Z033429-2 | 1 ug used |

*Continued on next page*

*Continued*

| Reagent type (species) or resource | Designation | Source or reference | Identifiers | Additional information |
|---|---|---|---|---|
| Peptide, recombinant protein | Protein A conjugated magnetic beads | Invitrogen/Thermo Fisher Sci | Cat# 10008D | |
| Peptide, recombinant protein | Protein G conjugated magnetic beads | Invitrogen/Thermo Fisher Sci | Cat# 10009D | |
| Peptide, recombinant protein | BSA Fraction V | Sigma-Aldrich | Cat# 10735094001 | |
| Peptide, recombinant protein | T4 ligase | New England Bio | Cat# M0202S | |
| Peptide, recombinant protein | Phusion DNA Polymerase | New England Bio | Cat# M0530L | |
| Commercial assay or kit | T7 Select 10-3b Cloning kit | EMD Millipore | Cat# 70550–3 | |
| Commercial assay or kit | Ampure XP Beads | Beckman Coulter | Cat# A63881 | |
| Software, algorithm | CD-HIT | *Fu et al., 2012*; *Li and Godzik, 2006* | http://weizhongli-lab.org/cd-hit/ | |
| Software, algorithm | numpy | Open Source | https://doi.org/10.1109/MCSE.2011.37 | |
| Software, algorithm | scipy | Open Source | https://www.nature.com/articles/s41592-019-0686-2 | |
| Software, algorithm | Matplotlib | Open Source | https://ieeexplore.ieee.org/document/4160265 | |
| Software, algorithm | Cutadapt | *Martin, 2011* | https://cutadapt.readthedocs.io/en/stable/ | |
| Software, algorithm | Cytoscape | *Shannon et al., 2003* | https://cytoscape.org | |

## Ethical approval

The study protocol was reviewed and approved by the Makerere University School of Medicine Research and Ethics Committee (Identification numbers 2011–149 and 2011–167), the London School of Hygiene and Tropical Medicine Ethics Committee (Identification numbers 5943 and 5944), the University of California, San Francisco, Committee on Human Research (Identification numbers 11–05539 and 11–05995) and the Uganda National Council for Science and Technology (Identification numbers HS-978 and HS-1019). Written informed consent was obtained from all participants in the study. For children, this was obtained from the parents or guardians.

## Study sites and participants

Plasma samples for the study were obtained from the Kanungu and Tororo sites of the Program for Resistance, Immunology, and Surveillance of Malaria (PRISM) cohort studies, part of the East African International Centers of Excellence in Malaria Research (*Kamya et al., 2015*). Kihihi sub-county in Kanungu district is a rural highland area in southwestern Uganda characterized by moderate transmission; samples used from this region were collected between 2012 and 2016. Nagongera sub-county in Tororo district is a rural area in southeastern Uganda with high transmission and samples used from this region were collected between Aug and Sep 2012. Samples from Tororo were restricted to individuals with fewer than six malaria cases per year to exclude individuals with very high incidence. Entomological inoculation rates (EIR) used in the study were calculated for each household based on entomological surveys involving collection of mosquitoes with CDC light traps and quantifying the number of *P. falciparum*-containing female anopheles mosquitoes along with sporozoite rates (*Kilama et al., 2014*). These cohorts and study sites have been described extensively in prior publications *Helb et al., 2015*; *Kamya et al., 2015*; *Rek et al., 2016*; *Yeka et al., 2015*; briefly, follow up consisted of continuous passive surveillance for malaria at a study clinic open 7 days a week where all routine medical care was provided, routine active surveillance for parasitemia, and routine entomologic surveillance. One plasma sample was selected from each of 100 participants, stratified by age, from each of the two cohorts. The 86 US controls were de-identified plasma obtained from adults who donated blood to the New York Blood Center.

## Bioinformatic construction of Falciparome Phage Library

The pipeline for library construction is shown in *Figure 1—figure supplement 1*. To construct the library, raw protein sequence files were downloaded from their respective public databases. Coding sequences from 3D7 and IT strains were downloaded from PlasmoDB (*Amos et al., 2022*) and vaccine/viral sequences were downloaded from the RefSeq database (*O'Leary et al., 2016*). Antigenic variant sequences were curated from multiple sources. The entire collection of protein sequences used as input for designing the peptides in the study can be found in the Dryad dataset doi:10.7272/Q69S1P9G. Pseudogenes were removed and any remaining stop codons within coding sequences were replaced with Alanine residues. These sequences were combined and filtered using CD-HIT (*Fu et al., 2012*; *Li and Godzik, 2006*) to remove sequences with >x% identity, where the threshold X used varied for different sets of sequences are in *Table 2*.

The final set of protein sequences (n=8980) was then merged and short sequences (<30 aa long) were removed prior to collapsing at 100% sequence identity (n=8534). Next, all sequences were split into 62-amino acid peptide fragments with 25-amino acid step size. Fragments with homopolymer runs of $\geq$ 8 exact amino acid matches in a row were removed, X amino acids were substituted to Alanine and Z amino acids (Glutamic acid or Glutamine) to Q (Glutamine), and finally, lzw compression was used to identify and remove low-complexity sequences with a compression ratio less than 0.4. Lastly, sequence headers were renamed to remove spaces and the resulting peptide fragments were converted to nucleotide sequences. Adapter sequences were appended, with a library-specific linker on the 5' end (GTGGTTGGTGCTGTAGGAGCA) and a 3' linker sequence coding for two stop codons and a 17mer (-TGATAA- GCATATGCCATGGCCTC). This file was then iteratively scanned for restriction enzyme sites (EcoRI, HindIII), which were eliminated by replacement with synonymous codons to facilitate cloning. The final set of nucleotide sequences was collapsed at 100% nucleotide sequence identity (n=238,068) and then ordered from Agilent Technologies.

## Cloning and packaging into T7 phage

A single vial of lyophilized DNA was received from Agilent. The lyophilized oligonucleotides were resuspended in 10 mM Tris–HCl-1 mM EDTA, pH 8.0 to a final concentration of 0.2 nM and PCR amplified for cloning into T7 phage vector arms (Novagen/EMD Millipore Inc T7 Select 10–3 Cloning kit). Detailed protocol can be found in 10.17504/protocols.io.j8nlkkrr5l5r/v1. Four 30 µl packaging reactions were performed and all were pooled in the end. Plaque assays were done with the packaging reaction to determine the titer of infectious phage in the packaging reaction and estimated to be $2\times10^8$ pfu/ml. Phage libraries were then prepared and amplified fresh from packaging reactions. Resulting phage libraries were tittered by plaque assay and adjusted to a working concentration of $10^{10}$ pfu/mL before incubation with patient plasma.

## Immunoprecipitation of antibody-bound phage

Plasma samples were first diluted in 1:1 storage buffer (0.04% NaN3, 40% Glycerol, 40 mM HEPES (pH 7.3), 1 x PBS (-Ca and –Mg)) to preserve antibody integrity. Then, a 1:2.5 x dilution of that stock was made in 1 x PBS resulting in a final 1:5 dilution and 1 µl of this was used in PhIP-seq. The protocol was followed as in 10.17504/protocols.io.j8nlkkrr5l5r/v1. Forty µl of Pierce Protein A/G Bead slurry (ThermoFisher Scientific) were used per sample. After round 1 of IP, the eluted phage were amplified in *E. coli* and enriched through a second round of IP. The final lysate was spun and stored at –20 °C for NGS library prep. Immunoprecipitated phage lysate was heated to 70 °C for 15 min to expose DNA. DNA was then amplified in two subsequent reactions. All samples had a minimum of two technical replicates.

## Bioinformatic analysis of PhIP-Seq data

### Identification of seroreactive peptides

Sequencing reads were first trimmed to cut out adaptors with Cutadapt (*Martin, 2011*):

Trimmed reads were then aligned to the full Falciparome peptide library using GSNAP (*Wu and Nacu, 2010*) paired end alignment, outputting a SAM file:

For each aligned sequence, the CIGAR string was examined, and all alignments where the CIGAR string did not indicate a perfect match were removed. The final set of peptides was tabulated to generate counts for each peptide in each individual sample. Samples with less than 250,000 aligned

reads were dropped from further analysis and any resulting samples with only one technical replicate were also dropped (2 of the 200 Ugandan samples were dropped). To keep the analysis restricted to *P. falciparum* peptides and limit the influence from non-*P. falciparum* peptides, reads mapping to all vaccine, viral and experimental control peptides were excluded from analysis. The remaining peptide counts were normalized for read depth and multiplied by 500,000, resulting in reads/500,000 total reads (RP5K) for each peptide. The null distribution for each peptide was modeled using read counts from a set of 86 plasma from the US (New York Blood Center) using a normal distribution, with the assumption that most of these individuals were likely unexposed to malaria. To avoid inflation by division, if the standard deviation of read counts of any peptide in the US samples was <1, then that was set as 1. Z-score enrichments ((x-mean US)/std. dev US) were then calculated for each peptide in each sample using the US distribution and Z-score ≥ 3 in both technical replicates (or more than 75% of the replicates if there were more than two technical replicates) of a sample was used to identify enriched peptides within a given sample. To call malaria-specific peptide enrichments ('seroreactive peptides'), enrichment was required in a minimum of five Ugandan samples. Seropositivity for a peptide was calculated by the percent of Ugandan samples enriching for that peptide. Seropositivity for a protein was calculated by the percent of Ugandan samples enriching for any peptide within that protein.

## Calculation of breadth of non-redundant peptide groups per person

Seroreactive peptides in each person were collapsed based on shared sequences (7-mer identical motifs) using the network approach described in AVARDA (*Monaco et al., 2021*) to get a conservative estimate of the number of non-redundant peptide groups in each person.

## Calculation of VSA breadth per person

VSA breadth was calculated as the number of variant proteins in each VSA family that were seroreactive in a given person and was calculated as follows. Since all these families possess conserved as well as variable regions, during library design, peptides across conserved regions from many variants that share identical sequences were filtered out to avoid redundant representation and only one representative peptide was retained in the final Falciparome library. Therefore, to accurately calculate the number of VSA proteins recognized in a person, all seroreactive VSA peptides were mapped back to the sequences from the full collection of VSA protein sequences to identify all the variant proteins each seroreactive peptide sequence mapped to. This information was then used to get the number of variant proteins reactive to a person's plasma. Domain classification for PfEMP1s was done using the VarDom server (*Rask et al., 2010*). Domain classification for RIFINs was done based on *Joannin et al., 2008*.

## Repeat analysis

Only unique 3D7/IT proteins in the library (if both 3D7 and IT homologs were present in the library, only the 3D7 homolog was considered) that were not members of Variant Surface Antigens (PfEMP1, RIFIN, STEVOR, SURFIN, Pfmc-2TM) were considered for all repeat analysis to avoid redundancy of representation.

*Cumulative repeat frequency in proteins* - For calculation of cumulative repeat frequency in proteins, amino acid sequences of proteins were input into the RADAR (*Madeira et al., 2019*) program for denovo identification of repeats using default settings. Cumulative frequency of repeats in a protein was then determined by adding the repeat counts of all reported repeats in the protein. To compare to the non-seroreactive set, the same number of proteins as the seroreactive protein set was randomly sampled from the total non-seroreactive protein set 1000 times and the distribution of cumulative frequencies between the seroreactive and non-seroreactive sets were compared using a 2-sample KS-test in each iteration.

*Repeat index calculation* - To systematically compare the distribution of repeats between seroreactive and non-seroreactive peptides within seroreactive proteins, the following approach was adopted. Firstly, for each protein, repeats and their frequency within that protein was calculated using a k-mer approach. K-mers were fixed length sequences (6/7/8/9-aa) with any number of conservative substitutions (AG, DE, RHK, ST, NQ, LVI, YFW) and did not include polymeric stretches of single amino acids from N/Q/D/E/R/H/K. For each protein sequence, all possible kmers in the protein and their frequency (number of non-overlapping occurrences) in the protein (intra-protein repeat frequency)

was calculated. Then for each peptide in the protein, all k-mers in the peptide sequence were taken and the k-mer with the highest intra-protein repeat frequency was identified. This frequency was assigned as the repeat index for the peptide. Once all peptides across all seroreactive proteins were assigned a repeat index, they were subsequently classified according to seropositivity. In each sero-positivity group, since peptides from the same protein could have the same highest intra-protein repeat k-mer, to avoid redundancy of representation, peptides sharing the same highest k-mer were collapsed and counted only once. For the non-seroreactive peptide set, random sampling of peptides from all non-seroreactive peptides was performed (1000 iterations). The 2-sample KS test was then used to compare distributions.

### Inter-protein motif analysis

First, all motifs with wildcards (any amino acid allowed at that position) or conservative substitutions (AG, DE, RHK, ST, NQ, LVI, YFW), shared by at least two seroreactive peptides were identified using the SLiMFinder program (*Edwards et al., 2007*), a part of the SLiMSuite package. The following parameters were used for running the program with the seroreactive peptide sequences as input: teiresias = T efilter = F blastf = F masking = F ftmask = F imask = F compmask = F metmask = F slimlen = 7 absmin = 2 absminamb = 2 slimchance = F maxwild = 1 maxseq = 10,000 walltime = 240 minocc = 0.0002 ambocc = 0.0003 wildvar = False equiv = <txt file that lists the allowed conservative substitutions - AG, DE, RHK, ST, NQ, LVI, YFW>.

Following this, a custom script was used to parse motifs with desired length and degeneracy threshold and identify those enriched over background. First, motifs of length K with at least N fixed positions and allowed number of conservative substitutions and wildcards were filtered depending on the degeneracy thresholds used. Motifs with homopolymeric stretches of KKKKK/ NNNNN/ EEEEE were not considered as this is a common feature in the proteome of plasmodium. Then, for each motif, the number of seroreactive peptides possessing that motif was determined (frequency in the seroreactive set). Next, enrichment in the seroreactive set over background was estimated with the following approach. Random sampling was performed on the whole library to get the same number of random peptides as seroreactive peptides (n=9927) and the occurrence frequency of each motif was calculated in the random set each time. This was bootstrapped 1000 times and this represented the background frequency of the motifs in 1000 iterations. A p-value for enrichment in the seroreactive set was then calculated using a Poisson model for the background frequency distribution. Significantly enriched ones were then identified following multiple hypothesis correction (FDR of 0.1%). This set of motifs represented the final collection of significantly enriched motifs. From this set, those that were shared by at least two seroreactive proteins were identified as inter-protein motifs. Network visualizations were performed with Cytoscape (*Shannon et al., 2003*). For the analysis on PfEMP1 with random set of peptides, all PfEMP1 peptides from the seroreactive set (n=3001) were combined with random peptides (n=6926) to a total of 9927 peptides. This was treated as the 'seroreactive' set and a similar analysis was performed to identify significantly enriched motifs in this set.

### Data availability

The data and sample metadata associated with this study can be accessed in the Dryad repository with the doi:10.7272/Q69S1P9G (https://datadryad.org/stash/share/YuYmQNKNvrWmoMX8n99wle_2bFyrtweAGclxYPHkPjY).

The code generated for the study is on GitHub and can be accessed at https://github.com/madhura-raghavan/phage-malaria-uganda200.git, (copy archived at swh:1:rev:b49a3ec15d86048dd570b-340487cee139fbeb445; *Raghavan, 2023*).

### Acknowledgements

We thank all study participants who participated in this study and their families. We thank the New York Blood Center for providing us with the de-identified human control plasma samples. We thank Caleigh Mandel-Brehm for advice and discussions on PhIP-seq. We thank members of the DeRisi and Greenhouse labs for helpful discussions.

Joseph L DeRisi - Chan Zuckerberg Biohub. Bryan Greenhouse - CZB Investigator program, NIH/NIAID awards A1089674 (East Africa ICEMR), AI119019, and AI144048. The funders had no role in study design, data collection and interpretation, or the decision to submit the work for publication.

## Additional information

### Competing interests

Isabel Rodriguez-Barraquer: Reviewing editor, *eLife*. Joseph L DeRisi: Paid scientific advisor for Allen & Co. Paid scientific advisor for the Public Health Company, Inc and holds stock options. Founder and holding stock options in VeriPhi Health, Inc. The other authors declare that no competing interests exist.

### Funding

| Funder | Grant reference number | Author |
|---|---|---|
| Chan Zuckerberg Biohub | | Joseph L DeRisi |
| National Institutes of Health | A1089674 (East Africa ICEMR) | Bryan Greenhouse |
| Chan Zuckerberg Biohub | Investigator program | Bryan Greenhouse |
| National Institutes of Health | AI119019 | Bryan Greenhouse |
| National Institutes of Health | AI144048 | Bryan Greenhouse |

The funders had no role in study design, data collection and interpretation, or the decision to submit the work for publication.

### Author contributions

Madhura Raghavan, Conceptualization, Data curation, Software, Formal analysis, Validation, Investigation, Visualization, Methodology, Writing – original draft, Writing – review and editing; Katrina L Kalantar, Software, Methodology, Writing – review and editing; Elias Duarte, Validation, Methodology; Noam Teyssier, Software, Formal analysis; Saki Takahashi, Chris Drakeley, Resources, Writing – review and editing; Andrew F Kung, Software, Writing – review and editing; Jayant V Rajan, Methodology; John Rek, Kevin KA Tetteh, Isaac Ssewanyana, Resources; Isabel Rodriguez-Barraquer, Conceptualization, Formal analysis, Investigation, Visualization, Writing – review and editing; Bryan Greenhouse, Conceptualization, Resources, Formal analysis, Supervision, Funding acquisition, Investigation, Visualization, Writing – original draft, Project administration, Writing – review and editing; Joseph L DeRisi, Conceptualization, Resources, Software, Formal analysis, Supervision, Funding acquisition, Investigation, Visualization, Methodology, Writing – original draft, Project administration, Writing – review and editing

### Author ORCIDs

Madhura Raghavan http://orcid.org/0000-0002-8566-9189
Elias Duarte http://orcid.org/0000-0003-2541-5504
Chris Drakeley http://orcid.org/0000-0003-4863-075X
Isabel Rodriguez-Barraquer http://orcid.org/0000-0001-6784-1021
Bryan Greenhouse http://orcid.org/0000-0003-0287-9111
Joseph L DeRisi http://orcid.org/0000-0002-4611-9205

### Ethics

The study protocol was reviewed and approved by the Makerere University School of Medicine Research and Ethics Committee (Identification numbers 2011-149 and 2011-167), the London School of Hygiene and Tropical Medicine Ethics Committee (Identification numbers 5943 and 5944), the University of California, San Francisco, Committee on Human Research (Identification numbers 11-05539 and 11-05995) and the Uganda National Council for Science and Technology (Identification numbers HS-978 and HS-1019). Written informed consent was obtained from all participants in the

study. For children, this was obtained from the parents or guardians. The US control samples were from New York Blood Center and these samples came from volunteer blood donors who consented as follows, "I authorize NYBC to use or transfer my blood or portions of it for any purpose it deems appropriate, including transfusion, research, or commercial purposes."

### Decision letter and Author response

Decision letter https://doi.org/10.7554/eLife.81401.sa1
Author response https://doi.org/10.7554/eLife.81401.sa2

## Additional files

### Supplementary files

- Supplementary file 1. List of 9927 seroreactive peptides identified in this dataset with their sequences.
- Supplementary file 2. Top 40 proteins with highest seropositivity and associated literature.
- Supplementary file 3. List of top 100 proteins with highest seropositivity used for GO analysis.
- Supplementary file 4. Seropositivity rate (proportion of people seropositive) for all 9927 seroreactive peptides across different groups in the two exposure settings.
- Supplementary file 5. Seropositivity rate (proportion of people seropositive) for top repeat elements across different groups in the two exposure settings.
- Supplementary file 6. List of inter-protein motifs and the proteins sharing them. Motifs reported here are 7-mers with at least 5 identical amino acids and up to two conservative substitutions (and no wildcards).
- Supplementary file 7. Table describing the number of interprotein motifs obtained with varied parameters for calling the motifs.
- Supplementary file 8. Gene network file for interprotein motifs (7-mers with at least 5 identical amino acids and up to two conservative substitutions (and no wildcards)). Can be visualized on Cytoscape.
- MDAR checklist

### Data availability

All data generated or analyzed during this study are included in the manuscript, supporting files and in the Dryad repository with the https://doi.org/10.7272/Q69S1P9G.

The following dataset was generated:

| Author(s) | Year | Dataset title | Dataset URL | Database and Identifier |
|---|---|---|---|---|
| Raghavan M, Kalantar K, Duarte E, Teyssier N, Takahashi S, Kung A, Rajan J, Rek J, Tetteh K, Drakeley C, Ssewanyana I, Rodriguez-Barraquer I, Greenhouse B, DeRisi J | 2022 | Proteome-wide antigenic profiling in Ugandan cohorts identifies associations between age, exposure intensity, and responses to repeat-containing antigens in *Plasmodium falciparum* | https://dx.doi.org/10.7272/Q69S1P9G | Dryad Digital Repository, 10.7272/Q69S1P9G |

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
