## [Editor Report]

Malaria immunity is complex, and this new platform, namely the phage display of *Plasmodium falciparum* proteome-wide peptides for profiling of antibody targets, provides a valuable addition to the toolkit for understanding humoral responses. The study, conducted using plasma from Ugandan children and adults, represents an important aspect of naturally acquired antibodies with sera-reactive responses to the intra-and inter-protein repeat regions. The revised version solidly supports the claims of the authors; it contains a reanalysis of cohort comparisons accounting for infection status, updated analyses of cross-reactive epitopes to account for within-individual effects, and it emphasizes the limitations in the conclusions.

---

## [Decision Letter]

**Decision letter after peer review:**

Thank you for submitting your article "Proteome-wide antigenic profiling in Ugandan cohorts identifies associations between age, exposure intensity, and responses to repeat-containing antigens in *Plasmodium falciparum*" for consideration by *eLife*. Your article has been reviewed by 3 peer reviewers, and the evaluation has been overseen by a Reviewing Editor and Dominique Soldati-Favre as the Senior Editor. The reviewers have opted to remain anonymous.

Essential revisions:

1) Please conduct analyses of the data stratified by BS/PCR status.

2) Please include analyses of antibody data to confirm your hypothesis that "inter-protein motifs" do cross-react.

3) Please revise the manuscript according to the additional suggestion indicated by the numbered lines corresponding to the text in the manuscript.

*Reviewer #1 (Recommendations for the authors):*

Specific comments

Interprotein motifs as potential cross-reactive epitopes:

Line 59: "short motifs associated with seroreactivity were extensively shared among hundreds of antigens, potentially representing cross-reactive epitopes." Please present analyses showing whether reactivity to these shared motifs indeed correlates between different proteins within the same individual. One presumes so but this can be assessed with available data.

Lines 396-479 – an extended passage is dedicated to the analysis of interprotein motifs, and their association with seroreactive proteins. The authors should analyze whether seroreactivities to different peptides containing the same/similar interprotein motif correlate or not within individuals. This would test the authors' hypothesis on L 399 "If these motifs are a part of an epitope, then antibodies and B-cell receptors (BCR) specific to a motif can potentially cross-react with the motif variants in different proteins".

L 608: "Another limitation of our study is that it did not provide quantitative measures of absolute antibody reactivity to individual peptides per person." This is a bit confusing, since Figure 2 —figure supplement 2 shows that "Technical replicates are well correlated" for seroreactivity to individual peptides, indicating the assay is quantitative (but perhaps not for "antibody reactivity", although that is the underlying selective force?). In any case, it should be possible to compare RP5K between peptides that share the same interprotein motif, assuming that these motifs represent important B cell epitopes as the authors hypothesize.

Effects of concurrent parasitemia on antibody:

Line 61: "PfEMP1 shared motifs with the greatest number of other antigens". This echoes prior studies showing short-term responses against different (heterologous) PfEMP1 variants during/after infection, which is an important potential confounder in the analyses presented here. The authors should address/analyze this issue of concurrent malaria infection effects more thoroughly, otherwise, any comparison between the two cohorts is hard to interpret.

Line 174 – please clarify if there was a rationale for the different characteristics of the sample sets between the two sites – "the majority of Tororo samples were positive for infection", while those in Kanungu were "100 days after their last infection". This will bias any comparisons between the two cohorts. For example, Helb et al. previously observed, with sera from these same study sites, that "Pf-Specific Antibody Profiles Showed Decreased Responses with Increased Days Since Infection."

Line 254 – "children in the higher transmission setting had a significantly higher breadth than children in the moderate transmission setting." Here and elsewhere in the Results, the analyses could be confounded by concurrent parasitemia during sampling (which is more common in the high transmission cohort), since acute parasitemia has acute effects on antibody profiles including cross-reactive antibodies (for example, see https://pubmed.ncbi.nlm.nih.gov/15175751/). The authors should compare individuals with patent parasitemia (by microscopy), subpatent parasitemia (by PCR), and without parasitemia at the time of sample collection, to understand these potential confounding effects.

L 532-536: "The difference in seroreactivity to repeat-containing peptides observed here between the [high versus moderate transmission] settings could therefore emerge from two related mechanisms. In the first, the difference could be driven by a requirement for a minimum level of cumulative exposure to the target repeats to generate a robust response. In the second, the antibody response to repeats may be inherently less durable, leading to rapid waning in the absence of frequent exposure." Please test the possibility that these differences among children are simply related to concurrent parasitemia at the time of sample collection.

Platform limitations:

Line 228 – "none of the proteins expressed in the mosquito oocyst stage were identified as seroreactive" Kindly clarify this statement, since for example CSP is expressed during the oocyst stage.

Line 239 – "a known limitation of PhIP-seq – it detects predominantly linear epitopes". The passage including this statement may leave readers with the impression that the previous protein array studies cited here (which appear to have all used the same in vitro protein translation system) were generated using properly folded proteins with conformationally correct epitopes. However, this may be unlikely given the difficulty to express properly folded malaria antigens. Many of the antigens targeted by protective immunity are highly folded proteins and functional antibodies against these antigens are frequently conformation-dependent. As such, one might expect the PhIP-seq platform (and the other protein array platforms they cite) to perform poorly at studying such antibodies. The authors should clarify these issues a bit more as context for readers. The limitation of this PhIP approach may be indicated by "Figure 2 —figure supplement 7 Breadth of seroreactivity in the variable regions of RIFIN and PfEMP1" which shows modest reactivity of adult sera to peptides of PfEMP1, whereas prior studies have established that adult sera in Africa cross-react to many/all blood-stage parasite surface variants circulating in a community (eg, see Figure 1 in https://pubmed.ncbi.nlm.nih.gov/10882604/ ), presumably reflecting seroreactivity to the immunodominant PfEMP1 proteins.

Line 267 – "In the moderate transmission setting, adults had a significantly higher breadth of PfEMP1 variants recognized than children, suggesting an age and/or cumulative exposure dependent increase in PfEMP1 breadth in this setting, as previously observed in (Cham et al., 2009)." Please provide numbers and results of statistical analysis to substantiate this statement. This difference looking at the figure is not striking-it appears reasonably clear for CIDR a/b/g, but less so for DBL domains.

Line 274: "a decline in responses to variants as children develop into adults in this [high transmission] setting. This is consistent with observations from a previous study investigating antibody responses to PfEMP1 DBLa domains in Papua New Guinea (Barry et al., 2011)". While the results are consistent, it's worth noting that Barry used an array of in vitro translated proteins that may not be properly folded-this was never tested in Barry's study and hence would have similar limitations to the study here that uses only linear epitopes.

Figure 1: please state in the figure the total number of falciparum proteins and corresponding peptides in the library-the number 8980 stated here includes >1400 non-malaria proteins (and these were apparently excluded from analysis-line 725).

*Reviewer #2 (Recommendations for the authors):*

Overall, this paper was excellent. The writing was careful, and the experiment is exciting.

"Slow development of immunity" is a motivation described in the introduction for investigating repeat-containing elements. I would recommend adding some nuance to this description of natural immunity to avoid confusion for some readers. Immunity to severe disease in children (and in pregnancy) is acquired quite rapidly after only one or two infections. This is relevant here because it may differentiate the immunity to the types of antigens focused on in this paper from variant antigens and genes with completely different, but discrete numbers, of alleles-e.g., MSP1, EBA175, etc.

A potential major confounder in the interpretation of results indicating responses to repeat elements that are shared amongst multiple proteins is that the assay wouldn't differentiate antibodies derived from one or many of these loci. The conclusions about the importance of inter-protein repeats on immunity are based on the assumption that any signal in these regions is based on antibodies derived from those proteins, rather than Ab's to one region causing the immunoprecipitation of the others. This seems especially relevant where seroreactivity is discussed. Is it possible to look at regions adjacent to the inter-protein repeats to discern this? I understand this was done when investigating seroreactivity within proteins, but this would be to confirm signals from different proteins. If it's just that I'm missing something obvious, some discussion clarifying why this isn't a confounder would be a good addition.

I thought the focus on repeat regions was important and wonderful to see, but some comparison to polymorphic loci feels like a lost opportunity to add significant value. It would be particularly interesting to know if an immunity to repeats was anticorrelated within individuals to other loci, and how this compares across transmission intensity. For example, does high repeat-element seroreactivity stymie development of antibodies to invasion receptors?

Another potential confounder would be if the parasites in one of your cohorts was more similar to 3D7/IT than the other. This might be assessed with a PCA plot of SNPs from your NGS reads, or perhaps with MalariaGen or PlasmoDB tools comparing different locations in Uganda.

*Reviewer #3 (Recommendations for the authors):*

Lines 218-220: "The 9,927 seroreactive peptides identified by the pipeline were derived from 1,648 parasite proteins and antigenic variants, many of which showed broad seroreactivity across pediatric and adult Ugandan samples"

The 9,927 seroreactive peptides were identified from 238,068 total peptides corresponding to 4.2% of the peptides. Similarly, 1,648 seroreactive proteins were identified from 8,980 total proteins corresponding to 18.4% of the proteins. Taken together, the seropositivity in this study was lower than the previous studies using protein array and α screen. Please discuss why this difference has happened.

The reader of this paper will be interested in the seropositivity ranking among the top 100 proteins (not only the list of Top protein names; Supplementary table 2). Please provide the ranking of the seropositive proteins and add a discussion about the selected proteins.

As I listed in the public review 1), the phage display system is the ability to sequentially enrich and amplify the signal to noise. Although more rounds of IP definitely increase the S/N, an increased number of rounds of IP will have disadvantages such as biased results. Please add a description of why the authors defined the methods to round 2 of IP, not 1 or 3 or more.

---

## [Author Response]

Essential revisions:1) Please conduct analyses of the data stratified by BS/PCR status.

We thank the reviewers for this suggestion. We have added this analysis, taking into account both exposure and infection status. (lines 409-435). Two new supplementary figures (Figure 5 Figure Supplement 3, Figure 5 Figure Supplement 4), also accompany the new text to support these analyses.

2) Please include analyses of antibody data to confirm your hypothesis that "inter-protein motifs" do cross-react.

We thank the reviewers for this suggestion. We have added this analysis to the manuscript (lines 495- 503). A new supplementary figure (Figure 6 Figure Supplement 3) also accompanies the new text to support this analysis.

3) Please revise the manuscript according to the additional suggestion indicated by the numbered lines corresponding to the text in the manuscript.

We have revised the manuscript as indicated by the numbered line suggestions.

Reviewer #1 (Recommendations for the authors):Specific commentsInterprotein motifs as potential cross-reactive epitopes:Line 59: "short motifs associated with seroreactivity were extensively shared among hundreds of antigens, potentially representing cross-reactive epitopes." Please present analyses showing whether reactivity to these shared motifs indeed correlates between different proteins within the same individual. One presumes so but this can be assessed with available data.

We have performed the suggested analysis. This has been added to the Results section (lines 495-503), as well as a new supplemental figure (Figure 6, Supplement 3) comparing seroreactivity of inter-protein motifs within individuals. We find significantly more sharing of seroreactivity to peptides with inter-protein motifs compared to peptides without inter-protein motifs, supporting the notion of cross-reactivity. Despite this association, we believe it prudent to be conservative about the conclusions, since the response of every individual is complex and polyclonal. Thus, we maintain our position that the data suggests wide-spread cross reactivity, but does not prove it. We have emphasized this point in the discussion.

Lines 396-479 – an extended passage is dedicated to the analysis of interprotein motifs, and their association with seroreactive proteins. The authors should analyze whether seroreactivities to different peptides containing the same/similar interprotein motif correlate or not within individuals. This would test the authors' hypothesis on L 399 "If these motifs are a part of an epitope, then antibodies and B-cell receptors (BCR) specific to a motif can potentially cross-react with the motif variants in different proteins".

As noted above, we have included a new analysis to address this question.

L 608: "Another limitation of our study is that it did not provide quantitative measures of absolute antibody reactivity to individual peptides per person." This is a bit confusing, since Figure 2 —figure supplement 2 shows that "Technical replicates are well correlated" for seroreactivity to individual peptides, indicating the assay is quantitative (but perhaps not for "antibody reactivity", although that is the underlying selective force?). In any case, it should be possible to compare RP5K between peptides that share the same interprotein motif, assuming that these motifs represent important B cell epitopes as the authors hypothesize.

While PhIPseq technical replicates are highly correlated, the assay is not considered quantitative in terms of titer or antibody affinity. As discussed in previous papers using PhIPseq ((Vazquez et al. 2020; Yuan et al. 2018), the actual number of reads recovered from an IP may vary due to many factors. This includes the starting copy number of phage targets in the library, position of the epitope in the peptide, non-specific interactions with the magnetic beads, and so on. In this work, and previous papers, the fold enrichment (by RPK) of any given peptide species is evaluated relative to the same library applied to control samples (non-malaria exposed plasma in this case), allowing a semi-quantitative, relative assessment of seroreactivity. An important aspect of this work is our inclusion of a large number (n=86) of (non-malaria) US controls, which forms the basis for measuring the relative enrichment in the Ugandan cohort. We have added a clarification in the limitations section of the discussion to hopefully avoid confusion (Lines 677-681).

As noted above, we have also included the requested analysis of per individual interprotein motif comparisons for cross-reactivity.

Effects of concurrent parasitemia on antibody:Line 61: "PfEMP1 shared motifs with the greatest number of other antigens". This echoes prior studies showing short-term responses against different (heterologous) PfEMP1 variants during/after infection, which is an important potential confounder in the analyses presented here. The authors should address/analyze this issue of concurrent malaria infection effects more thoroughly, otherwise, any comparison between the two cohorts is hard to interpret.

We thank the reviewer for this comment. The analysis on inter-protein motifs was processed using the entire set of 9927 seroreactive peptides, regardless of cohort, to obtain a landscape of potential cross-reactive sequences. Note that the quoted statement refers to a comparison of responses to PfEMP1 to those of other, non-PfEMP1 proteins, mitigating the potential effect of broad responses to heterologous PfEMP1 variants. Therefore, the result on motif sharing with PfEMP1 only suggests that with the widely diverse PfEMP1 variants, there is a potential for cross-reactivity with different non-PfEMP1 antigens depending on the variant expressed within an individual. We claim only that the potential for cross reactivity exists, by sequence similarity alone, among the seroreactive peptide sequences. We have made the limitations of this claim clear in the discussion (Line 632-635).

We have however tested the important confounder of concurrent malaria with respect to breadth of reactivity to repeats as suggested by the reviewer, as described in the public review.

Line 174 – please clarify if there was a rationale for the different characteristics of the sample sets between the two sites – "the majority of Tororo samples were positive for infection", while those in Kanungu were "100 days after their last infection". This will bias any comparisons between the two cohorts. For example, Helb et al. previously observed, with sera from these same study sites, that "Pf-Specific Antibody Profiles Showed Decreased Responses with Increased Days Since Infection."

The samples were selected based on age stratification and not based on positivity for infection per se – the descriptive characteristics of the sample sets mentioned by the reviewer partially reflect the fact that the individuals in Tororo were much more often infected and thus many more will be infected at the time of sampling.

Line 254 – "children in the higher transmission setting had a significantly higher breadth than children in the moderate transmission setting." Here and elsewhere in the Results, the analyses could be confounded by concurrent parasitemia during sampling (which is more common in the high transmission cohort), since acute parasitemia has acute effects on antibody profiles including cross-reactive antibodies (for example, see https://pubmed.ncbi.nlm.nih.gov/15175751/). The authors should compare individuals with patent parasitemia (by microscopy), subpatent parasitemia (by PCR), and without parasitemia at the time of sample collection, to understand these potential confounding effects.

As suggested by the reviewer and described in public review, we have added new analyses with respect to current and recent parasitemia. The practical limitations in sample size with this study, did not allow us to stratify by patent/subpatent parasitimea however.

L 532-536: "The difference in seroreactivity to repeat-containing peptides observed here between the [high versus moderate transmission] settings could therefore emerge from two related mechanisms. In the first, the difference could be driven by a requirement for a minimum level of cumulative exposure to the target repeats to generate a robust response. In the second, the antibody response to repeats may be inherently less durable, leading to rapid waning in the absence of frequent exposure." Please test the possibility that these differences among children are simply related to concurrent parasitemia at the time of sample collection.

This is included in our new analyses, as described above.

Platform limitations:Line 228 – "none of the proteins expressed in the mosquito oocyst stage were identified as seroreactive" Kindly clarify this statement, since for example CSP is expressed during the oocyst stage.

This is correct. Our original analysis was based on mass spectrometry data deposited in PlasmoDB until 2019, and using a spectral count >= 2 for calling expression, CSP was not called as expressed in the oocyst stage. Since then, newer datasets have been deposited and we have re-analyzed using all the currently available datasets, and we reduced the threshold to spectral count >= 1 to improve sensitivity, at the potential cost of specificity. Using this revised analysis, we now observe some oocyst sporozoite proteins (including CSP) identified in the seroreactive set. Notably though, the number of oocyst-only proteins is small, consistent with less representation of mosquito stage-specific proteins in the seroreactive set. We have updated the manuscript accordingly (lines 231-233 and Figure 2c).

Line 239 – "a known limitation of PhIP-seq – it detects predominantly linear epitopes". The passage including this statement may leave readers with the impression that the previous protein array studies cited here (which appear to have all used the same in vitro protein translation system) were generated using properly folded proteins with conformationally correct epitopes. However, this may be unlikely given the difficulty to express properly folded malaria antigens. Many of the antigens targeted by protective immunity are highly folded proteins and functional antibodies against these antigens are frequently conformation-dependent. As such, one might expect the PhIP-seq platform (and the other protein array platforms they cite) to perform poorly at studying such antibodies. The authors should clarify these issues a bit more as context for readers. The limitation of this PhIP approach may be indicated by "Figure 2 —figure supplement 7 Breadth of seroreactivity in the variable regions of RIFIN and PfEMP1" which shows modest reactivity of adult sera to peptides of PfEMP1, whereas prior studies have established that adult sera in Africa cross-react to many/all blood-stage parasite surface variants circulating in a community (eg, see Figure 1 in https://pubmed.ncbi.nlm.nih.gov/10882604/ ), presumably reflecting seroreactivity to the immunodominant PfEMP1 proteins.

We agree and thank the reviewer for pointing out this important point that the previous protein array studies may suffer from similar limitations of PhIP-seq in not displaying the conformations properly, especially for malaria proteins. We have modified the text to convey this point (lines 247).

With regards to PfEMP1, we acknowledge that our findings could just indicate a decrease in breadth of variant PfEMP1 recognition through linear epitopes in adults, and perhaps there is an increase in breadth of recognition through conformational epitopes that cannot be captured in our assay. This point has been added in lines 293-295.

Line 267 – "In the moderate transmission setting, adults had a significantly higher breadth of PfEMP1 variants recognized than children, suggesting an age and/or cumulative exposure dependent increase in PfEMP1 breadth in this setting, as previously observed in (Cham et al., 2009)." Please provide numbers and results of statistical analysis to substantiate this statement. This difference looking at the figure is not striking-it appears reasonably clear for CIDR a/b/g, but less so for DBL domains.

We have included this analysis in the Results section as suggested by the reviewer (lines 273-274) and we added additional detail to Figure 2, Supplement 6, including the ATS domain. In the moderate setting, the median number of PfEMP1 recognized in adults was 44, whereas the median in children was 26. For the moderate setting, the majority of seroreactivity was localized to the ATS domain, which is unsurprising, since this domain is conserved. The variable domains are inherently unlikely to be perfectly reflected between the PhIPseq library and the actual domains present in parasites from this cohort, which may result in less overall sensitivity in this assay. The text has been amended to clarify this point (lines 276-279).

Line 274: "a decline in responses to variants as children develop into adults in this [high transmission] setting. This is consistent with observations from a previous study investigating antibody responses to PfEMP1 DBLa domains in Papua New Guinea (Barry et al., 2011)". While the results are consistent, it's worth noting that Barry used an array of in vitro translated proteins that may not be properly folded-this was never tested in Barry's study and hence would have similar limitations to the study here that uses only linear epitopes.

Yes, as noted above, we have pointed out this limitation in the revision.

Figure 1: please state in the figure the total number of falciparum proteins and corresponding peptides in the library-the number 8980 stated here includes >1400 non-malaria proteins (and these were apparently excluded from analysis-line 725).

We have changed Figure 1 now to represent the total number of malaria proteins and peptides instead of overall total and added this to the text (lines 189-190).

Reviewer #2 (Recommendations for the authors):Overall, this paper was excellent. The writing was careful, and the experiment is exciting.

We thank the reviewer for these comments.

"Slow development of immunity" is a motivation described in the introduction for investigating repeat-containing elements. I would recommend adding some nuance to this description of natural immunity to avoid confusion for some readers. Immunity to severe disease in children (and in pregnancy) is acquired quite rapidly after only one or two infections. This is relevant here because it may differentiate the immunity to the types of antigens focused on in this paper from variant antigens and genes with completely different, but discrete numbers, of alleles-e.g., MSP1, EBA175, etc.

Agreed. We have added a modification to the Introduction, as suggested by the reviewer (lines 101-102).

A potential major confounder in the interpretation of results indicating responses to repeat elements that are shared amongst multiple proteins is that the assay wouldn't differentiate antibodies derived from one or many of these loci. The conclusions about the importance of inter-protein repeats on immunity are based on the assumption that any signal in these regions is based on antibodies derived from those proteins, rather than Ab's to one region causing the immunoprecipitation of the others. This seems especially relevant where seroreactivity is discussed. Is it possible to look at regions adjacent to the inter-protein repeats to discern this? I understand this was done when investigating seroreactivity within proteins, but this would be to confirm signals from different proteins. If it's just that I'm missing something obvious, some discussion clarifying why this isn't a confounder would be a good addition.

Yes, thank you. There are two points here. First, we did indeed examine the regions immediately adjacent to the inter-protein repeat motifs, which is possible since our library consists of overlapping peptides, giving us an effective resolution of 12-13 amino acids (Figure 6, Supplement 2). Here, we observed that seroreactivity is indeed localized to the part of the peptide with the interprotein motif, as opposed to other parts of the peptide. Second, we have added a new analysis of cross-reactivity between shared interprotein motifs within each individual (see Lines 495-503, and response to Reviewers #1 and #2). There are limitations to this analysis, which we have made clear in the manuscript as well (Lines 632-635).

I thought the focus on repeat regions was important and wonderful to see, but some comparison to polymorphic loci feels like a lost opportunity to add significant value. It would be particularly interesting to know if an immunity to repeats was anticorrelated within individuals to other loci, and how this compares across transmission intensity. For example, does high repeat-element seroreactivity stymie development of antibodies to invasion receptors?

This is an interesting question, although difficult to directly address with this dataset. The question is whether repeats stymie or interfere with the development of immune responses to particular classes of proteins (like invasion proteins). As a whole, we observe more seroreactivity to both repeats and non-repeats together as a function of exposure and age (Author response image 1). That said, individuals vary widely with respect to seropositivity to specific repeats and non-repeats, and therefore it would require a much larger dataset to establish co-occurrence, or interference, since no two individuals are alike. This is graphically depicted in Figure 2, Supplement 2 (top panel). The Ugandan samples are clearly more correlated than the US samples, due to malaria exposure, but high correlation values between individuals are lacking. Furthermore, interference may be relieved in the form of antibody feedback – for instance, it has been shown that feedback from antibodies to the immunodominant repeat epitopes in CSP can lead to diversification of responses to subdominant epitopes in a vaccine setting (Mcnamara et al. 2020). A correlation analysis using this dataset is unlikely to capture these nuances.

**Author response image 1. sa2fig1:** 

However, within proteins, it is clear that the linear peptide regions that are immuno-dominant are the repeats themselves (Figure 3, and also Figure 4), which is consistent with the notion that repeats prevent or shield against the development of seroreactivity to the non-repeat portions of the same protein. This effect has been posited in ((Schofield 1991)) and elsewhere. As we’ve pointed out, one limitation on this conclusion is that PhIPseq detects largely non-conformational epitopes.

Another potential confounder would be if the parasites in one of your cohorts was more similar to 3D7/IT than the other. This might be assessed with a PCA plot of SNPs from your NGS reads, or perhaps with MalariaGen or PlasmoDB tools comparing different locations in Uganda.

In this dataset, the NGS reads are from our phage library, and not from Ugandan parasite sequences themselves, and thus we do not have the direct data to answer this question. However, parasite diversity in Uganda (and sub-saharan Africa in general) is amongst the highest in the world, with little population structure within Uganda, as we have previously shown (Chang et al. 2017)

We thus expect any differences in genetic composition of parasites between the sites overall to be quite small in comparison to the extensive variation within each site and the large differences between parasites to which people are exposed and the reference parasites.

Reviewer #3 (Recommendations for the authors):Lines 218-220: "The 9,927 seroreactive peptides identified by the pipeline were derived from 1,648 parasite proteins and antigenic variants, many of which showed broad seroreactivity across pediatric and adult Ugandan samples"The 9,927 seroreactive peptides were identified from 238,068 total peptides corresponding to 4.2% of the peptides. Similarly, 1,648 seroreactive proteins were identified from 8,980 total proteins corresponding to 18.4% of the proteins. Taken together, the seropositivity in this study was lower than the previous studies using protein array and α screen. Please discuss why this difference has happened.

We thank the reviewer for this comment. We would like to point out that the 238,068 peptides and 8,980 proteins in the library represent the 3D7 and IT proteomes along with variant proteins as well as viral and experimental control proteins, as described in Table 2 (the complete peptide list and sequences are also available at Dryad doi:10.7272/Q69S1P9G). The set of 1,648 seroreactive proteins identified here represent unique proteins, to avoid double counting from 3D7/IT or variant antigens. Thus, these 1,648 proteins represent ~30% of the 5,400 proteome *of P. falciparum*, and we have clarified this in the Results (Lines 221-222).

We note that we implemented a highly conservative inclusion criteria for proteins in this set to minimize false positives (Figure 2, Supplement 4) by leveraging the large number of US control samples and enforcing a minimum level of 5 individuals sharing. We note that previous studies either did not have a large control (unexposed) set and/or did not require sharing among at least 5 individuals. The comparison of the 1,648 proteins identified here with previous screens are discussed in detail in both the Results (Lines 239-247) and Discussion section (Lines 565-571).

The reader of this paper will be interested in the seropositivity ranking among the top 100 proteins (not only the list of Top protein names; Supplementary table 2). Please provide the ranking of the seropositive proteins and add a discussion about the selected proteins.

Yes, agreed. A “top ranking” summary is now provided (Supplementary table 2b), in addition to the full dataset.

As I listed in the public review 1), the phage display system is the ability to sequentially enrich and amplify the signal to noise. Although more rounds of IP definitely increase the S/N, an increased number of rounds of IP will have disadvantages such as biased results. Please add a description of why the authors defined the methods to round 2 of IP, not 1 or 3 or more.

We and others have measured the gain in signal as a function of IP round (O’Donovan et al., “High-resolution epitope mapping of anti-Hu and anti-Yo autoimmunity by programmable page display. *Brain Communications* 2020). As shown in the above reference, multiple rounds amplifies the signal. Pilot experiments on Ugandan samples done with 1, 2 or 3 rounds using the Falciparome library showed that the largest fold-gain in S/N for known antibody targets in *P. falciparum* was realized between round 1 and 2, and thus we proceeded with 2 rounds to avoid potential bias from phage amplification.

References

Chang, Hsiao Han, Colin J. Worby, Adoke Yeka, Joaniter Nankabirwa, Moses R. Kamya, Sarah G. Staedke, Grant Dorsey, et al. 2017. “THE REAL McCOIL: A Method for the Concurrent Estimation of the Complexity of Infection and SNP Allele Frequency for Malaria Parasites.” PLoS Computational Biology 13 (1). https://doi.org/10.1371/JOURNAL.PCBI.1005348.

Mcnamara, Hayley A, Azza H Idris, Henry J Sutton, Mattia Bonsignori, Robert A Seder, Ian A Cockburn Correspondence, Rachel Vistein, et al. 2020. “Antibody Feedback Limits the Expansion of B Cell Responses to Malaria Vaccination but Drives Diversification of the Humoral Response.” https://doi.org/10.1016/j.chom.2020.07.001.

Schofield, L. 1991. “On the Function of Repetitive Domains in Protein Antigens of Plasmodium and Other Eukaryotic Parasites.” Parasitology Today (Personal Ed.) 7 (5): 99–105. https://doi.org/10.1016/0169-4758(91)90166-L.

Vazquez, Sara E., Elise M.N. Ferré, David W. Scheel, Sara Sunshine, Brenda Miao, Caleigh Mandel-Brehm, Zoe Quandt, et al. 2020. “Identification of Novel, Clinically Correlated Autoantigens in the Monogenic Autoimmune Syndrome APS1 by Proteome-Wide Phip-Seq.” *eLife* 9 (May). https://doi.org/10.7554/ELIFE.55053.

Yuan, Tiezheng, Divya Mohan, Uri Laserson, Ingo Ruczinski, Alan N Baer, H Benjamin Larman, and Corresponding. 2018. “Improved Analysis of Phage ImmunoPrecipitation Sequencing (PhIP-Seq) Data Using a Z-Score Algorithm.” BioRxiv, April, 285916. https://doi.org/10.1101/285916.